Manuscript prepared for Geosci. Model Dev.
with version 2014/09/16 7.15 Copernicus papers of the LaTeX class copernicus.cls.
Date: 27 August 2017

# An axisymmetric non-hydrostatic model for double-diffusive water systems

Koen Hilgersom[1], Marcel Zijlema[2], and Nick van de Giesen[1]

[1]Water Resources Section, Faculty of Civil Engineering and Geosciences, Delft University of Technology, Delft, P.O. Box 5048, 2600 GA, The Netherlands
[2]Environmental Fluid Mechanics Section, Faculty of Civil Engineering and Geosciences, Delft University of Technology, Delft, P.O. Box 5048, 2600 GA, The Netherlands

*Correspondence to:* Koen Hilgersom (k.p.hilgersom@tudelft.nl)

**Abstract.** The three-dimensional (3-D) modelling of water systems involving double-diffusive processes is challenging due to the large computation times required to solve the flow and transport of constituents. In 3-D systems that approach axisymmetry around a central location, computation times can be reduced by applying a 2-D axisymmetric model set-up. This article applies the Reynolds-averaged Navier-Stokes equations described in cylindrical coordinates, and integrates them to guarantee mass and momentum conservation. The discretized equations are presented in a way that a Cartesian finite volume model can be easily extended to the developed framework, which is demonstrated by the implementation into a non-hydrostatic free-surface flow model. This model employs temperature and salinity dependent densities, molecular diffusivities, and kinematic viscosity. One quantitative case study, based on an analytical solution derived for the radial expansion of a dense water layer, and two qualitative case studies demonstrate a good behaviour of the model for seepage inflows with contrasting salinities and temperatures. Four case studies with respect to double-diffusive processes in a stratified water body demonstrate that turbulent flows are not yet correctly modelled near the interfaces, and that an advanced turbulence model is required.

## 1 Introduction

Over the past decades, numerical salt and heat transport models have increased their capability to capture patterns of double-diffusion on scales varying from laboratory set-ups to the ocean (Yoshida and Nagashima, 2003; Kunze, 2003). Despite the advance in computation power and parallel computing, the requirement of dense grids for the three-dimensional (3-D) modelling of salt and heat transport often yields unacceptable computation times. In this article, we present a framework for

a finite volume approach that allows free-surface flow modelling in a 2-D axisymmetric grid. The model framework is intended for a shallow water body where salinity and temperature gradients potentially induce double-diffusive processes. As such, the model intends to simulate larger-scale features of double-diffusion (i.e., interface locations in a stratified system and heat and salt trans-

port).

Kunze (2003) stresses that numerical and analytical methods to model double-diffusion often only apply at specific scales. For example in oceans, internal wave-shear and strain enhance salt-finger growth, leading to higher salt and heat fluxes over stratified interfaces. Traxler et al. (2011) addresses the issue of scale by describing four modes of instability in salt-fingering systems, which play a role

on different scales. Their 3-D simulations of large-scale instability in salt-fingering systems are the first known successful direct numerical simulations (DNS). Carpenter et al. (2012) were among the first to model systems in the double-diffusive convection regime with 3-D DNS. Their detailed simulations showed that, in this regime, the salt and heat fluxes across the interface are largely governed by molecular diffusion and that these salt and heat diffusion rates control the thickness

of the salt and heat interface, respectively. Kimura and Smyth (2007) used 3-D DNS to model salt sheets for a double-diffusively stratified flow interacting with inflectional shear.

Yoshida and Nagashima (2003) have shown that 2-D numerical models are already well able to simulate small-scale processes in laboratory set-ups. On a larger scale, Sommer et al. (2014) confirm the findings of Carpenter et al. (2012) with 2-D DNS and high-resolution measurements of a double-

diffusive staircase in Lake Kivu for density ratios larger than 3, noting that in these systems external turbulence by shear or internal waves should be absent to maintain diffusion as the main driver for salt and heat transport. Noguchi and Niino (2010a, b) used 2-D DNS to study the spontaneous layer formation in the double-diffusive convection regime and explores the layer formation from the non-linear evolution of disturbances.

Most numerical modelling studies of double-diffusive processes calculate interfaces and salt and heat fluxes at oceanic scale (Stommel and Fedorov, 1967; Stern, 1967; Ruddick and Gargett, 2003; Kelley et al., 2003; Kunze, 2003; Kimura et al., 2011). This can be explained by the ubiquity of these systems in oceans (Huppert and Turner, 1981), and by the potential of oceanic thermohaline strati-fication as an energy source (Stommel et al., 1956; Vega, 2002). These larger-scale simulations are

commonly performed with Reynolds-averaged Navier-Stokes (RANS) models. For example, Radko et al. (2014a, b) successfully applied a 3-D RANS model to an oceanic scale salt-finger staircase. Recently, modelling of these phenomena in smaller-scale water bodies has started to be developed. For example, double-diffusive processes like thermohaline staircasing have been successfully mod-elled in lakes (Schmid et al., 2003), although these systems are generally modelled with analytical

or empirical formulations (Kelley et al., 2003; Schmid et al., 2004; Arnon et al., 2014). Other known numerical modelling studies consider double-diffusive convection in monitoring wells (Berthold and Börner, 2008), and the collection of thermal energy in solar ponds (Cathcart and Wheaton, 1987; Gi-

estas et al., 2009; Suárez et al., 2010, 2014). However, modelling these complex physical processes in shallow waters still imposes a major scientific and computational challenge (Dias and Lopes, 60  2006).

Axisymmetric CFD models are applied in a wide variety of fields. Examples of applications include the modelling of flow of gas past a gravitating body in astronomy (Shima et al., 1985), radiative heat transfer in cylindrical enclosures (Menguc and Viskanta, 1986), the heating of air flowing through a combustion burner (Galletti et al., 2007), and acoustic axisymmetric waves in elastic media 65  (Schubert et al., 1998). The similarity between these examples is that a model calculating in two spatial dimensions models 3-D processes due to axisymmetry. In geohydrology, axisymmetric models are often applied for groundwater flow around injection and abstraction wells (Bennett et al., 1990). Groundwater modelling software often offers code extensions that adjust several input parameters to allow such modelling approaches (Reilly and Harbaugh, 1993; Langevin, 2008).

In some cases, axisymmetric grid set-ups can also be preferential for hydrodynamic surface water models. Examples of such cases are close-to-circular water bodies with uniform boundaries, and the flow around a central point (e.g., a local inflow from a pipe or groundwater seepage). The occurrence of local saline seepage inflows into shallow water bodies of contrasting temperatures has been described by De Louw et al. (2013). Hilgersom et al. (2016) has shown how these local inflows can 75  induce thermohaline stratification in the shallow surface water bodies above these inflows.

This article derives a framework for an axisymmetric free-surface RANS model, which is implemented in SWASH. SWASH is an open source non-hydrostatic modelling code for the simulation of coastal flows including baroclinic forcing (SWASH source code, 2011). It is suitable for the simulation of flows and transport in varying density fields, because 1) the staggered grid allows a mo-80  mentum and mass conservative solution of the governing equations, which is required for accurate salt and heat transport modelling, and 2) the non-hydrostatic pressure terms aid the simulation of flows in fields with large density variations. Another major advantage of SWASH is the flexible and easily extendible code, which can be applied for free under the GNU GPL license. Other properties of SWASH are the opportunity to apply terrain-following $\sigma$-layers for the definition of cell depths 85  and the user-friendly pre- and post-processing.

The development of an axisymmetric variation of SWASH falls in line with our research to localized saline water seepage in Dutch polders. To simulate the effect of a local seepage inflow on the temperature profile of the surface water body, a numerical model is required that accounts for sharp density gradients, a free surface and potential double-diffusive processes. The axisymmetric 90  grid set-up aids in correctly representing the volumetric inflow and modelling the flow processes around the local inflow.

In this article, we present the resulting numerical framework to extend a 2-D finite volume model into a 2-D axisymmetric model by adding few terms to the solution of the governing Navier-Stokes and transport equations. These terms are implemented in the SWASH code. The model code is

95  further extended with a new transport module calculating salt and heat transfer. Although the model generally calculates with a mesh size that is larger than the size required to solve small-scale double-diffusive instabilities, the aim is to allow the model to approximate interface locations and salt and heat fluxes. The functioning of the code is validated with case studies involving different salinity and temperature gradients.

## 2  Method

### 2.1  Governing equations

The governing equations in this study are the Reynolds-averaged Navier-Stokes equations for the flow of an incompressible fluid, derived in cylindrical coordinates $(r, \alpha, z)$ (Batchelor, 1967). Due to point symmetry, the gradients in tangential direction $(\alpha)$ are set to zero, which leaves the solution of the equations in one horizontal and one vertical dimension (i.e., 2-DV):

$$\frac{1}{r}\frac{\partial ur}{\partial r} + \frac{\partial w}{\partial z} = 0 \tag{1}$$

$$\frac{\partial u}{\partial t} + \frac{\partial uu}{\partial r} + \frac{\partial wu}{\partial z} = -\frac{1}{\rho}\frac{\partial p}{\partial r} + \left( \frac{1}{r}\frac{\partial}{\partial r}\left( \nu_h r \frac{\partial u}{\partial r}\right) - \frac{\nu_h u}{r^2} + \frac{\partial}{\partial z}\left( \nu_v \frac{\partial u}{\partial z}\right)\right) \tag{2}$$

$$\frac{\partial w}{\partial t} + \frac{\partial uw}{\partial r} + \frac{\partial ww}{\partial z} = -\frac{1}{\rho}\frac{\partial p}{\partial z} + \frac{1}{r}\frac{\partial}{\partial r}\left( \nu_h r \frac{\partial w}{\partial r}\right) + \frac{\partial}{\partial z}\left( \nu_v \frac{\partial w}{\partial z}\right) - g \tag{3}$$

In these equations, $r$ represents the horizontal axis in radial direction and $z$ the vertical axis, with $u$ and $w$ the velocities along these axes, respectively. The density $\rho$ is calculated from the local temperature and salinity states by the updated Eckart formula (Eckart, 1958; Wright, 1997), which is based on the UNESCO IES80 formula (Unesco, 1981).

This RANS model allows turbulence modelling with the standard k-$\epsilon$ model (Launder and Spalding, 1974). This article presents cases that are modelled with and without this turbulence model. In case of the former, the modelled eddy viscosity is added to the molecular viscosity, yielding a non-uniform vertical viscosity $\nu_v$. For all the calculations, the horizontal kinematic viscosity $\nu_h$ is set uniform to its molecular value ($\sim 10^{-6}$ m$^2$s$^{-1}$).

The pressure terms are split into hydrostatic and hydrodynamic terms, according to Casulli and Stelling (1998):

$$\frac{1}{\rho}\frac{\partial p}{\partial r} = \frac{g}{\rho}\frac{\partial \int_{z'=z}^{\zeta} \rho\left(r, z', t\right) dz'}{\partial r} + \frac{\partial q}{\partial r} \tag{4}$$

$$\frac{1}{\rho}\frac{\partial p}{\partial z} + g \equiv \frac{\partial q}{\partial z} \tag{5}$$

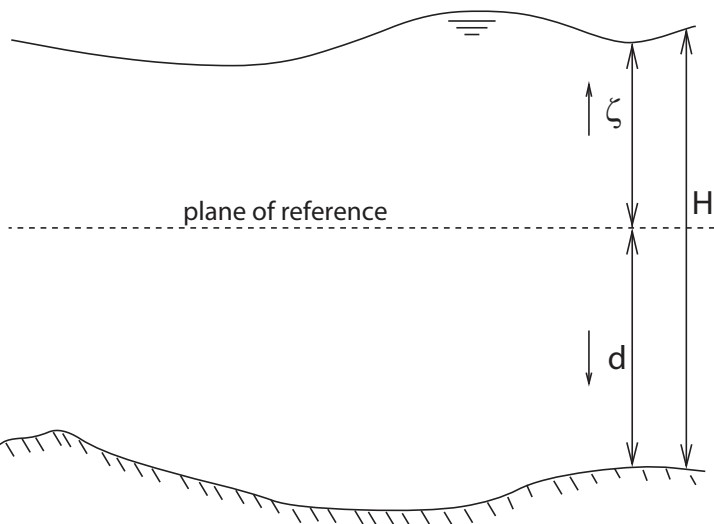

**Figure 1.** Definition of the free surface level $\zeta$ and the bottom level $d$ (Zijlema and Stelling, 2005).

where $q$ denotes the hydrodynamic pressure component and $\zeta$ the local free surface level relative to the reference plane (Fig. 1). Horizontal variations in atmospheric pressure are neglected. The first right-hand side term of Eq. 4 is split into baroclinic and barotropic components when the equations are integrated over the cell depth in Section 2.3. In the vertical, the baroclinic pressure gradient and the gravitational acceleration cancel each other out, leaving the hydrodynamic pressure gradient
(Equation 5).

     The free surface is calculated according to Zijlema and Stelling (2008), by integrating Eq. 1 over the depth of the water column and applying the free surface condition $w|_{z=\zeta} = \partial\zeta/\partial t + u\partial\zeta/\partial r$:

$$\frac{\partial \zeta}{\partial t} + \frac{1}{r}\frac{\partial r\bar{Q}}{\partial r} = 0$$
$$\bar{Q} \equiv UH = \int_{-d}^{\varsigma} u\, dz \tag{6}$$

     where $U$ is the depth-averaged velocity, and $d$ is the local bottom depth (Fig. 1). $\bar{Q}$ represents the
radial discharge per unit tangential width.

     Transport of mass and heat is calculated with the convection-diffusion equation:

$$\frac{\partial c}{\partial t} + \frac{1}{r}\frac{\partial ruc}{\partial r} + \frac{\partial wc}{\partial z} = \frac{1}{r}\frac{\partial}{\partial r}\left(D_h r \frac{\partial c}{\partial r}\right) + \frac{\partial}{\partial z}\left(D_v \frac{\partial c}{\partial r}\right) \tag{7}$$

     where the concentration $c$ represents either the salinity $S$ or temperature $T$.

     In the case that turbulence is modelled, the vertical turbulent diffusion, $D_v$, is calculated by adding
the molecular diffusivity and turbulent diffusivity: $D_v = D_{mol} + D_{turb}$. The turbulent diffusivity is

calculated by dividing the eddy viscosity $\nu_{turb}$ by the turbulent Prandtl number ($Pr = 0.85$) in the case of heat transport, or by the turbulent Schmidt number ($Sc = 0.7$) in the case of salt transport:

$$D_{turb;T} = \frac{\nu_{turb}}{Pr} = \frac{\nu_{turb}}{0.85} \tag{8}$$

$$D_{turb;S} = \frac{\nu_{turb}}{Sc} = \frac{\nu_{turb}}{0.7} \tag{9}$$

with $D_{turb;T}$ and $D_{turb;S}$ being the thermal and solutal turbulent diffusivities in $\mathrm{m^2 s^{-1}}$, respectively.

In non-turbulent thermohaline systems, stability largely depends on density gradients and molecular heat and salt diffusion rates, which in turn are highly dependent on temperature and salinity. The heat and salt diffusivities are related to temperature $T$ (°C) and salinity $S$ (weight $-$ %) by a quadratic regression on data presented in the International Critical Tables of Numerical Data,

Physics, Chemistry and Technology (Washburn and West, 1933):

$$D_{mol;T} = 1.31721 + 4.26657 \cdot 10^{-3} \cdot T - 1.09237 \cdot 10^{-6} \cdot T^2 + 1.74051 \cdot 10^{-2} \cdot S -$$
$$3.17759 \cdot 10^{-4} \cdot S^2 \tag{10}$$

$$D_{mol;S} = 7.66025 + 2.33023 \cdot 10^{-1} \cdot T + 3.21974 \cdot 10^{-3} \cdot T^2 - 2.18290 \cdot 10^{-1} \cdot S +$$
$$1.34431 \cdot 10^{-2} \cdot S^2 \tag{11}$$

## 2.2 Boundary conditions

At the free surface, we assume no wind and $q|_{z=\zeta} = 0$. At the bottom boundary, the vertical velocity

is calculated by imposing the kinematic condition $w|_{z=-d} = -u \partial d / \partial r$. The presented case studies (Section 3) include a local seepage inflow at the bottom boundary, for which the seepage velocity is added to the kinematic condition. For horizontal momentum, the bottom friction is imposed by applying a constant friction coefficient to the bottom layer, or by the logarithmic wall law in case the standard k-$\epsilon$ model is employed, applying a Nikuradse roughness height to determine the amount of

friction (Launder and Spalding, 1974).

A special case is the inner boundary, where symmetry occurs: for all variables, the gradient is set to zero, except for horizontal momentum: $u|_{r=0} = 0$. For the presented case studies, we define a Dirichlet boundary condition for $u$ momentum at the outer boundary, where the total outflow is equated to the instantaneous seepage inflow.

For the transport equation, a homogeneous Neumann boundary condition is defined at each boundary ($\frac{\partial cr}{\partial r} = 0$ and $\frac{\partial c}{\partial z} = 0$), except at a defined seepage inflow of known temperature and salt concentration, where a Dirichlet boundary condition is imposed.

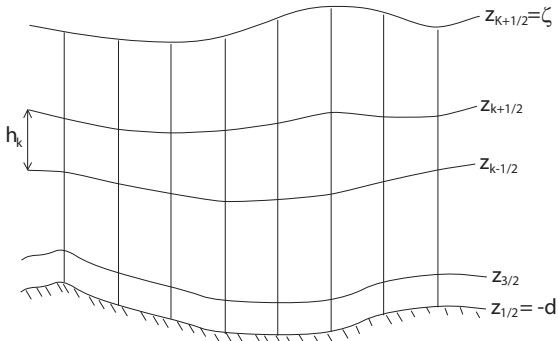

**Figure 2.** Vertical grid definition (sigma layers) (Zijlema and Stelling, 2005)

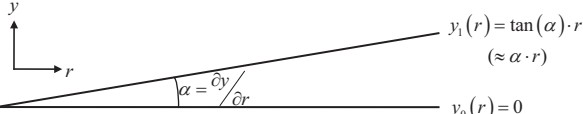

**Figure 3.** Axis definition

### 2.3 Numerical framework and implementation

The physical domain is discretized with a fixed cell width in radial direction. The width of the cells
in the tangential direction increases by a fixed angle $\alpha$, which allows us to consider the horizontal
grid as a pie slice (Fig. 3). In the model, $\alpha$ could be assigned any value (i.e., also $2\pi$ for a completely
circular grid). However, to allow a simple presentation of the integration step in this subsection, we
consider $\alpha$ as a small angle.

For the vertical grid, sigma layering is employed, although part of the layers can be defined by
a fixed cell depth (Fig. 2). A classical staggered grid is applied with velocities defined at the cell
boundaries and the other states in the cell centre.

For reasons of momentum and mass conservation, Zijlema and Stelling (2005) integrated the
governing equations over the cell depth using the Leibniz integral rule (Appendix A). In our case,
the cell width in tangential direction varies as well. Therefore, the equations are integrated over the
cell depth and the width in tangential direction, which is in this case defined as the $y$-dimension. For
the continuity equation, this yields (cf. Fig. 2 and 3):

$$\int\limits_{z_{k-\frac{1}{2}}}^{z_{k+\frac{1}{2}}} \int\limits_{y_0}^{y_1} \left( \frac{1}{r}\frac{\partial ur}{\partial r} + \frac{\partial w}{\partial z} \right) dydz = \frac{\partial \phi_k y_1}{\partial r} + y_1\omega_{k+\frac{1}{2}} - y_1\omega_{k-\frac{1}{2}} + y_1\frac{\partial h_k}{\partial t} = 0 \tag{12}$$

with $\phi_k = u_k \cdot h_k$ is the cell depth integrated velocity and the relative vertical velocity $\omega$ as defined
in Eq. 16 of Zijlema and Stelling (2005).

The momentum equations and the transport equation are integrated in a similar fashion:

$$\overline{y_1}^r \frac{\partial u_k \overline{h_k}^r}{\partial t} + \frac{\partial \overline{\phi_k}^r \hat{u}_k \overline{y_1}^r}{\partial r} + \overrightarrow{y_1} \hat{u}_{k+\frac{1}{2}} \overline{\omega_{k+\frac{1}{2}}}^r - \overrightarrow{y_1} \hat{u}_{k-\frac{1}{2}} \overline{\omega_{k-\frac{1}{2}}}^r - \boxed{\alpha \phi_k u_k} + g\overline{h_k}^r \overline{y_1}^r \frac{\partial \zeta}{\partial r} +$$

$$\frac{\partial q_k h_k y_1}{\partial r} - \overline{y_1}^r \overline{q}^{rz} \frac{\partial z_{i+\frac{1}{2},k+\frac{1}{2}}}{\partial r} + \overline{y_1}^r \overline{q}^{rz} \frac{\partial z_{i+\frac{1}{2},k-\frac{1}{2}}}{\partial r} - \boxed{\alpha \overline{h_k}^r \overline{q_k}^r} + \overline{y_1}^r \frac{g}{\overline{\rho_k}} \frac{\partial \overline{\rho_k}}{\partial r} \frac{\left(\overline{h_k}^r\right)^2}{2} +$$

$$\frac{g\overline{h_k}^r \overline{y_1}^r}{\overline{\rho_k}} \sum_{j=1}^{k-1} \left( \overline{h_j}^r \frac{\partial \overline{\rho_j}}{\partial r} + (\overline{\rho_j} - \overline{\rho_k}) \frac{\partial h_j}{\partial r} \right) - \frac{\partial}{\partial r} \nu_h h_k y_1 \frac{\partial \overline{u_k}}{\partial r} +$$

$$\boxed{\alpha h_k u_k \frac{\nu_h}{r}} - \overline{y_1}^r \left[ \nu_v \frac{\partial u}{\partial z} \right]_{k-\frac{1}{2}}^{k+\frac{1}{2}} = 0 \quad (13)$$

$$y_1 \frac{\partial w_{k+\frac{1}{2}} \overline{h_{k+\frac{1}{2}}}^z}{\partial t} + \frac{\partial \hat{w}_{k+\frac{1}{2}} \overline{\phi_{k+\frac{1}{2}}}^z y_1}{\partial r} - \boxed{\alpha \overline{\phi_{k+\frac{1}{2}}}^{rz} w_{k+\frac{1}{2}}} + y_1 \hat{w}_{k+1} \overline{\omega_{k+1}}^z - y_1 \hat{w}_k \overline{\omega_k}^z +$$

$$y_1 q_{k+1} - y_1 q_k - \frac{\partial}{\partial r} \nu_h \overline{y_1}^r \overline{h_{k+\frac{1}{2}}}^{rz} \frac{\partial \overline{w}}{\partial r} - y_1 \left[ \nu_v \frac{\partial w}{\partial z} \right]_k^{k+1} = 0 \quad (14)$$

$$y_1 \frac{\partial c_k h_k}{\partial t} + \frac{\partial \phi_k c_k y_1}{\partial r} + y_1 \omega_{k+\frac{1}{2}} \hat{c}_{k+\frac{1}{2}} - y_1 \omega_{k-\frac{1}{2}} \hat{c}_{k-\frac{1}{2}} - \frac{\partial}{\partial r'} \left\{ D_h \overline{y_1}^r \overline{h_k}^r \frac{\partial c}{\partial r'} \right\} -$$

$$y_1 \left[ D_v \frac{\partial c}{\partial z'} \right]_{z_{k-\frac{1}{2}}}^{z_{k+\frac{1}{2}}} + \frac{\partial}{\partial r'} \left\{ D_h \overline{y_1}^r \overline{h_k}^r \frac{\partial \overline{c_k}^r}{\partial z'} \frac{\partial \overline{z}^z}{\partial r'} \right\} + y_1 \left[ D_h \frac{\partial z_k}{\partial r} \frac{\partial c}{\partial r'} \right]_{z_{k-\frac{1}{2}}}^{z_{k+\frac{1}{2}}} -$$

$$y_1 \left[ D_h \left( \frac{\partial z}{\partial r} \right)^2 \frac{\partial c}{\partial z'} \right]_{z_{k-\frac{1}{2}}}^{z_{k+\frac{1}{2}}} = 0 \quad (15)$$

where overlined variables denote spatially averaged values for these variables in $r$ or $z$ directions, and arrows denote the use of values from downstream cells. The boxes mark the *alpha terms*, which are the additional angular terms compared to the 2-DV solutions for the momentum equations in Cartesian coordinates. In the integrated transport equation (Eq. 15), the latter three terms on the left-hand side are the so-called anti-creepage terms, which should be incorporated for the calculation of transport when large gradients in water depth occur.

Since $u$ and $w$ are the primitive variables in the momentum equations, and not $uh$ and $wh$ as in Eq. 13 and 14, we further rewrite the momentum equations according to Zijlema and Stelling (2008). In order to do this for the $u$ momentum equation, we first spatially discretize the continuity equation in point $i + \frac{1}{2}$:

$$\overline{y_{1;i+\frac{1}{2}}}^r \frac{\partial \overline{h}_{i+\frac{1}{2},k}^r}{\partial t} + \frac{y_{1;i+1} \overline{\phi}_{i+1,k}^r - y_{1;i} \overline{\phi}_{i,k}^r}{\Delta r} + \overline{y}_{1;i+\frac{1}{2}}^r \left( \overline{\omega}_{i+\frac{1}{2},k+\frac{1}{2}}^r - \overline{\omega}_{i+\frac{1}{2},k-\frac{1}{2}}^r \right) = 0 \quad (16)$$

We then spatially discretize the $u$ momentum equation and expand $\partial u_k h_k / \partial t$ to $h_k \partial u_k / \partial t +$ $u_k \partial h_k / \partial t$. The latter term falls out by subtracting Eq. 16 multiplied by $u_k$ from Eq. 13:

$$
\overline{y}^r_{1;i+\frac{1}{2}} \overline{h}^r_{i+\frac{1}{2},k} \frac{\partial u_{i+\frac{1}{2},k}}{\partial t} + \frac{\overline{\phi}^r_{i+1,k} \overline{y}^r_{1;i+1} \left( \hat{u}_{i+1,k} - u_{i+\frac{1}{2},k} \right) - \overline{\phi}^r_{i+1,k} \overline{y}^r_{1;i+1} \left( \hat{u}_{i+1,k} - u_{i+\frac{1}{2},k} \right)}{\Delta r} +
$$
$$
\overrightarrow{y}_{1;i+\frac{1}{2}} \hat{u}_{i+\frac{1}{2},k+\frac{1}{2}} \overline{\omega}^r_{i+\frac{1}{2},k+\frac{1}{2}} - \overrightarrow{y}_{1;i+\frac{1}{2}} \hat{u}_{i+\frac{1}{2},k-\frac{1}{2}} \overline{\omega}^r_{i+\frac{1}{2},k-\frac{1}{2}} - \boxed{\alpha \phi_{i+\frac{1}{2},k} u_{i+\frac{1}{2},k}} +
$$
$$
g \overline{h}^r_{i+\frac{1}{2},k} \overline{y}^r_{1;i+\frac{1}{2}} \frac{\zeta_{i+1} - \zeta_i}{\Delta r} + \frac{q_{i+1,k} h_{i+1,k} y_{1;i+1} - q_{i,k} h_{i,k} y_{1;i}}{\Delta r} -
$$
$$
\overline{y}^r_{1;i+\frac{1}{2}} \overline{q}^{rz}_{i+\frac{1}{2},k+\frac{1}{2}} \frac{z_{i+1,k+\frac{1}{2}} - z_{i,k+\frac{1}{2}}}{\Delta r} + \overline{y}^r_{1;i+\frac{1}{2}} \overline{q}^{rz}_{i+\frac{1}{2},k-\frac{1}{2}} \frac{z_{i+1,k-\frac{1}{2}} - z_{i,k-\frac{1}{2}}}{\Delta r} -
$$
$$
\boxed{\alpha \overline{h}^r_{i+\frac{1}{2},k} \overline{q}^r_{i+\frac{1}{2},k}} + \overline{y}^r_{1;i+\frac{1}{2}} \frac{g}{\overline{\rho}^r_{i+\frac{1}{2},k}} \frac{\rho_{i+1,k} - \rho_{i,k}}{\Delta r} \frac{\left( \overline{h}^r_{i+\frac{1}{2},k} \right)^2}{2} +
$$
$$
\frac{g \overline{h}^r_{i+\frac{1}{2},k} \overline{y}^r_{1;i+\frac{1}{2}}}{\overline{\rho}^r_{i+\frac{1}{2},k}} \sum_{j=1}^{k-1} \left( \overline{h}^r_{i+\frac{1}{2},j} \frac{\rho_{i+1,j} - \rho_{i,j}}{\Delta r} + \left( \overline{\rho}^r_{i+\frac{1}{2},j} - \overline{\rho}^r_{i+\frac{1}{2},k} \right) \frac{h_{i+1,j} - h_{i,j}}{\Delta r} \right) -
$$
$$
\frac{\nu_h}{\Delta r} \left( h_{i+1,k} y_{1;i+1} \frac{u_{i+\frac{3}{2},k} - u_{i+\frac{1}{2},k}}{\Delta r} - h_{i,k} y_{1;i} \frac{u_{i+\frac{1}{2},k} - u_{i-\frac{1}{2},k}}{\Delta r} \right) + \boxed{\alpha \overline{h}^r_{i+\frac{1}{2},k} u_{i+\frac{1}{2},k} \frac{\nu_h}{r_{i+\frac{1}{2}}}} -
$$
$$
\overline{y}^r_{1;i+\frac{1}{2}} \left( \nu_{v;i+\frac{1}{2},k+\frac{1}{2}} \frac{u_{i+\frac{1}{2},k+1} - u_{i+\frac{1}{2},k}}{\overline{h}^{rz}_{i+\frac{1}{2},k+\frac{1}{2}}} - \nu_{v;i+\frac{1}{2},k-\frac{1}{2}} \frac{u_{i+\frac{1}{2},k} - u_{i+\frac{1}{2},k-1}}{\overline{h}^{rz}_{i+\frac{1}{2},k-\frac{1}{2}}} \right) = 0
$$
(17)

Again, the alpha terms are marked with boxes. Another addition compared to the Cartesian 2-DV solution are the $y$-factors throughout the equation, which serve as width compensation factors. For $w$ momentum, a similar procedure is applied.

    The governing equations are spatially discretized with a central differences approach, except for the advective terms. The advective terms are discretized with higher-order flux limiters (Fringer

et al., 2005), namely MINMOD flux limiters in the case of the momentum equations, and MUSCL flux limiters in the case of the transport equation.

    The horizontal time integration of the momentum and transport equations is Euler explicit. The horizontal advective terms in the momentum equations are solved with the predictor-corrector scheme of MacCormack (Hirsch, 1988). The vertical time integration is semi-implicit, applying the $\theta$-scheme.

The global continuity equation (Eq. 6) and barotropic forcing are solved semi-implicitly (Casulli and Cheng, 1992). The case studies (Section 2.4) apply an implicitness factor $\theta = 1$ (i.e., the Euler implicit scheme) for the vertical momentum and transport equations, the global continuity equation, and the barotropic forcing. The non-hydrostatic pressure is standard solved with the Euler implicit scheme. The complete discretizations are shown in Appendix B.

The numerical framework largely follows the SWASH solution procedure (Zijlema et al., 2011). The code was extended by adding the alpha terms and factors accounting for the varying cell width

**Table 1.** The dimensions, properties, and consequent stability parameters applied in the case studies. *Up* and *Down* refer to the upper and lower layer of the dual layered system (in Case 5 to 7, the lower temperatures and salinities are properties of the central inflow).

| Case | Dimension (m) | | $T$ (°C) | | $S$ (weight $-$ ‰) | | $w_{in}$ (ms$^{-1}$) | $Tu$ (°) | $R_\rho$ $-$ |
|------|------|------|------|------|------|------|------|------|------|
| | Depth | Radial | Up | Down | Up | Down | | | |
| 1 | 0.7 | 3.0 | 10 | 20 | 0 | 15 | - | -53.3 | 0.15 |
| 2 | 0.7 | 3.0 | 10 | 34 | 0 | 15 | - | -71.6 | 0.50 |
| 3 | 0.7 | 3.0 | 20 | 10 | 1 | 0 | - | 71.2 | 2.04 |
| 4 | 0.7 | 3.0 | 20 | 15 | 1 | 0 | - | 85.0 | 1.19 |
| 5 | 0.4 | 3.0 | 30 | 5 | 0 | 10 | $5 \cdot 10^{-4}$ | -13.2 | -0.62 |
| 6 | 0.5 | 1.5 | 20 | 25 | 1 | 3 | $1 \cdot 10^{-3}$ | -82.5 | 0.77 |
| 7 | 0.5 | 1.5 | 20 | 26 | 1 | 2.5 | $1 \cdot 10^{-3}$ | -96.4 | 1.25 |

in tangential direction. The density and transport calculation modules were replaced by new modules based on the selected density equation (Wright, 1997), and the presented diffusivity equations.

## 2.4 Verification and validation

This article validates the model qualitatively and quantitatively. The behaviour of a local seepage inflow setting on double-diffusive layering is verified qualitatively (Section 2.4.3). The quantitative validation tests the model results against:

1. documented properties of systems of double-diffusive convection and salt-fingering (Section 2.4.1);

2. the expected expansion of an unconditionally stable layer near a bottom seepage inflow, for which an analytical solution is derived in Section 2.4.2.

In all the case studies (Table 1), we applied a time step of 2 ms and a horizontal mesh size of 5 mm in radial direction. The vertical mesh size in Case 1 to Case 4 was set uniformly to 10 mm. In the Cases 5 to 7, the vertical mesh size varied over depth. Because the processes of most interest 235 occurred near the bottom, the mesh size was decreasing towards the bottom (Fig. 4).

### 2.4.1 Validation for double-diffusive characteristics

This subsection lists several common metrics, which we applied to quantitatively validate our simulations of double-diffusive systems with varying density gradients (Cases 1 to 4). To validate the applicability of the standard k-$\epsilon$ model, we present model simulations for each of these cases both 240 with and without the use of the turbulence model (Sections 3.1 and 3.2).

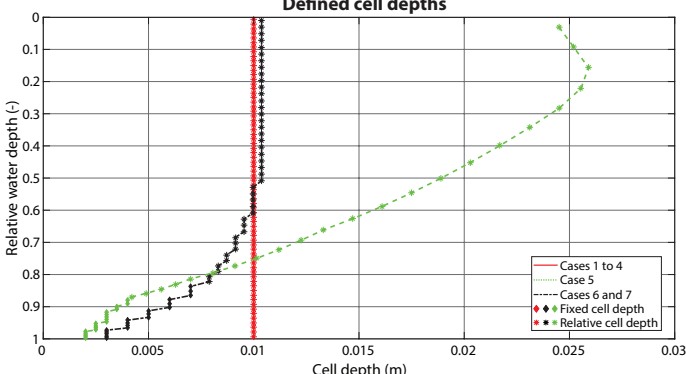

**Figure 4.** Defined cell depths for the Cases 1 to 7. For plotting reasons, the vertical axis displays the depth from the water surface relative to the local water depth. The cell depths that are defined relative to the local water depth (as marked by *) are displayed for the average water depth in each case study.

The stability of a double-diffusive system is commonly expressed by its Turner angle $Tu$ (Ruddick, 1983):

$$Tu = \arctan\left(\frac{N_T^2 - N_S^2}{N_T^2 + N_S^2}\right) \tag{18}$$

where the four quadrant arctangent function preserves the sign of the density gradients, $N_T^2 = -g \cdot \alpha_V \cdot \partial T/\partial z$, and $N_S^2 = g \cdot \beta_V \cdot \partial S/\partial z$. $\alpha_V$ (°C$^{-1}$) and $\beta_V$ ($10^3$ kg kg$^{-1}$) are the volumetric expansion coefficients for temperature and salinity, respectively, and the $z$-axis is in downward direction. A stable system occurs for $|Tu| < 45°$, whereas $|Tu| > 90°$ yields a gravitationally unstable system. Double-diffusive convection occurs for $-90° < Tu < -45°$, and salt-fingering for $45° < Tu < 90°$.

The expansion coefficients $\alpha_V$ and $\beta_V$ are varying with temperature and salinity itself, and are calculated for the average salinity and temperature on the interface. We stress, however, that the calculation of density gradients is highly sensitive to the assumed values of $\alpha_V$ and $\beta_V$. The dependencies of the expansion coefficients on temperature and salinity ($T$ (°C) and $S$ ($10^{-3}$ kg kg$^{-1}$), respectively) are derived from a linear regression to the density derivatives to $T$ and $S$, where the density is calculated according to Wright (1997):

$$\alpha_V(T,S) = -2.289087 \cdot 10^{-5} + 1.324960 \cdot 10^{-5} \cdot T - 9.289557 \cdot 10^{-8} \cdot T^2 +$$
$$1.563400 \cdot 10^{-6} \cdot S \tag{19}$$

$$\beta_V(T,S) = 7.999302 \cdot 10^{-4} - 2.777361 \cdot 10^{-6} \cdot T + 3.190719 \cdot 10^{-8} \cdot T^2 -$$
$$4.156012 \cdot 10^{-7} \cdot S \tag{20}$$

The Cases 1 and 2 concern a system with two layers of equal depth, where a cold and fresh water layer is overlying a warm and saline water layer (Table 1). Based on the Turner angle, double-diffusive convection is expected to occur. The onset of convection and salt and heat transport across the interface is induced by applying a few very small perturbations of order $10^{-6}$ °C throughout the temperature field. Case 1 has a smaller density ratio $R_\rho = -N_T^2/N_S^2$ than Case 2. Note that articles concerning double-diffusive convection commonly define $R_\rho$ as its reciprocal, in contrast to the common density gradient calculations for salt-finger systems. For the sake of consistency, this article employs one definition of the density ratio (i.e., the thermal density gradient over the saline density gradient).

Based on the Turner angles of Case 3 and Case 4, where warm and saline water is overlying cold and fresh water, salt-fingers are expected to occur (Table 1). Similar to the previous cases, we slightly perturbed the temperature field on a few locations. Case 3 has a larger density ratio than Case 4, yielding a lower salt flux over the interface (Kunze, 2003).

An effective transport of heat and salt over the interface while maintaining a sharp interface is expected as this is a known property of double-diffusive salt-fingers (Turner, 1965). Care should be taken that these salt-fingers are calculated in a 2-D radial grid. Yoshida and Nagashima (2003) pointed out that there is still a lack of knowledge about the 2-D and 3-D structures of salt-fingers and its implications for the interpretation of 2-D numerical results.

A clear definition of the interface location is relevant for the determination of the boundary properties and the heat and salt flux across the boundary. In each simulation, the interface location $z_{int}$ is defined for each depth profile of $S$ and $T$ as the location of the isoscalar. The isoscalar is constant and defined as the average value of $S$ and $T$ across the initial interface. Fig. 5 marks the interface locations for Case 3 and Case 4 as the locations of the isoscalars for times $t = 0$ h and $t = 6$ h.

The vertical saline and thermal density fluxes across the interface, $F_c$, are calculated on each grid location by time differentiating the salt and heat volumes above the interface according to Carpenter et al. (2012):

$$F_c = \frac{\mathrm{d}}{\mathrm{d}t} \int_0^{z_{int}} \rho_c(z)\,\mathrm{d}z \qquad (21)$$

where $\rho_c$ is the converted value of $S$ or $T$ to density units (i.e., $\rho_0\beta S$ or $\rho_0\alpha T$, $\rho_0$ being the reference density for the average salinity and temperature at the interface).

The simulated salt and heat fluxes are compared with theoretical fluxes based on molecular diffusivities and double-diffusion specific eddy diffusivities according to the equation (Carpenter et al., 2012):

$$F_{c;theoretical} = D \left.\frac{\mathrm{d}\rho_c}{\mathrm{d}z}\right|_{z_{int}} \qquad (22)$$

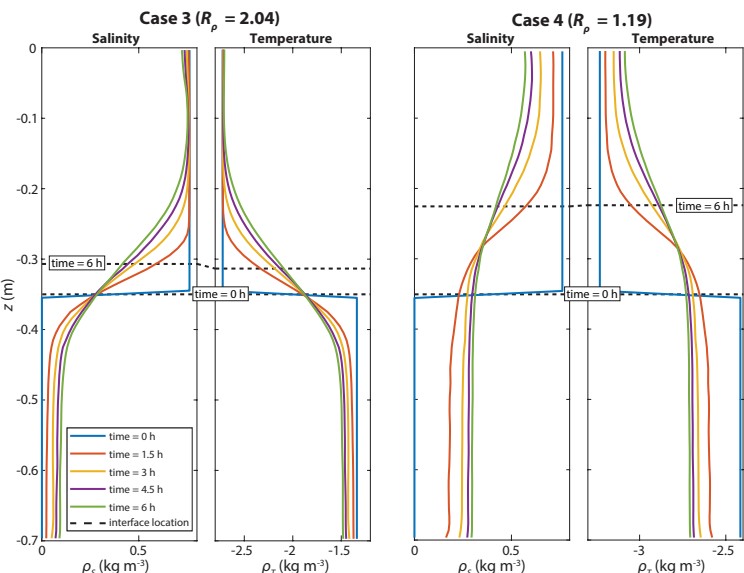

**Figure 5.** Interface positioning over time, displayed on the depth profiles of $S$ and $T$ (both in $\mathrm{kg m}^{-3}$ relative to the reference density), for the density ratios $R_\rho = 2.04$ (Case 3) and $R_\rho = 1.19$ (Case 4). The depth profiles are averaged over the complete horizontal domain and represent the simulations without a turbulence model.

with the derivative taken at each location of the isoscalar. Since this location is usually not located on the horizontal cell boundary, the derivative is determined by applying a weighted average to the derivatives at the neighbouring cell boundaries. $D$ can be either the molecular diffusivity $D_{mol}$ (Eq. 10 and Eq. 11) or an eddy diffusivity. In this article, the eddy diffusivities are only calculated
295  for salt diffusion by applying the following relationship with the molecular thermal diffusivity:

$$D_{eddy;S} = D_{mol;T} \frac{R_\rho}{\gamma} \tag{23}$$

with $\gamma = F_T \,/\, F_S$ being ratio of the heat and salt fluxes. A large variety of theoretical equations have been proposed for the flux ratio $\gamma$, both for salt-fingers (e.g., Stern, 1975):

$$\gamma_{Stern} = R_\rho - (R_\rho (R_\rho - 1))^{\frac{1}{2}} \tag{24}$$

300  and for double-diffusive convection (e.g., Kelley , 1990; Fernando, 1989):

$$\gamma_{Kelley} = \left( \frac{R_\rho^{-1} + 1.4 \left(R_\rho^{-1} - 1\right)^{\frac{3}{2}}}{1 + 14 \left(R_\rho^{-1} - 1\right)^{\frac{3}{2}}} \right)^{-1} \tag{25}$$

$$\gamma_{Fernando} = \tau^{\frac{1}{2}} R_\rho \tag{26}$$

where the Lewis number $\tau = D_{mol;T} / D_{mol;S}$ is the ratio of the molecular thermal and saline diffusivities.

For double-diffusive convection, we compare the heat fluxes also with theoretical heat fluxes as predicted by Kelley (1990) and Linden and Shirtcliffe (1978):

$$F_T^{Kelley} = 0.0032 \exp\left(\frac{4.8}{R_\rho^{-0.72}}\right) \left(\frac{gD_{mol;T}^2}{\rho_0 \nu}\right)^{1/3} \rho_T^{4/3} \tag{27}$$

$$F_T^{Linden \ \& \ Shirtcliffe} = \frac{1}{(\pi Ra_c)^{\frac{1}{3}}} \frac{\left(1 - \tau^{-\frac{1}{2}} R_\rho^{-1}\right)^{\frac{4}{3}}}{\left(1 - \tau^{-\frac{1}{2}}\right)^{\frac{1}{3}}} \left(\frac{gD_{mol;T}^2}{\rho_0 \nu}\right)^{1/3} \rho_T^{4/3} \tag{28}$$

with the critical Rayleigh number set to $Ra_c = 10^3$. In line with the common practice in this field of study, the heat fluxes are presented as a ratio to the heat flux through a solid plane (Turner, 1973):

$$F_T^{SP} = 0.085 \left(\frac{gD_{mol;T}^2}{\rho_0 \nu}\right)^{1/3} \rho_T^{4/3} \tag{29}$$

For systems of double-diffusive convection, we calculate the evolution of the boundary layer thicknesses $h_c$ according to Carpenter et al. (2012):

$$h_c = \frac{\Delta \rho_c}{\left.\frac{\partial \rho_c}{\partial z}\right|_{z_{int}}} \tag{30}$$

where the density difference between the upper and lower layer, $\Delta \rho_c$, is determined for averaged values of $c$ over the upper and lower quarter of its depth profile. The ratio of boundary layer thicknesses $r$ scales to $\tau$ by the relation $r \sim \tau^{\frac{1}{5}}$ and is expected to approach 2.5 for salt-heat systems (Carpenter et al., 2012).

As a last validation metric for the salt-finger cases, we employ the Stern number (Stern, 1969):

$$St = \frac{F_T - F_S}{\nu \left(\frac{\partial \rho_T}{\partial z} - \frac{\partial \rho_S}{\partial z}\right)} \tag{31}$$

Stern suggested that the growth of salt-fingers is arrested when $St$ reaches $O(1)$. However, Stern numbers have been reported varying from $O(10^{-3})$ to $O(10^2)$ for finger systems. Recently, Traxler et al. (2011) reported Stern numbers $St = 9.4$ and $St = 76$ for DNS simulations with density ratios $R_\rho = 2.0$ and $R_\rho = 1.2$. These density ratios are comparable to Case 3 and Case 4, allowing to compare our results with these DNS simulations.

### 2.4.2 Analytical validation of a stable inflow

The quantitative validation of an unconditionally stable bottom layer is based on an analytical solution for the radial expansion this dense layer from a central inflow under laminar flow conditions

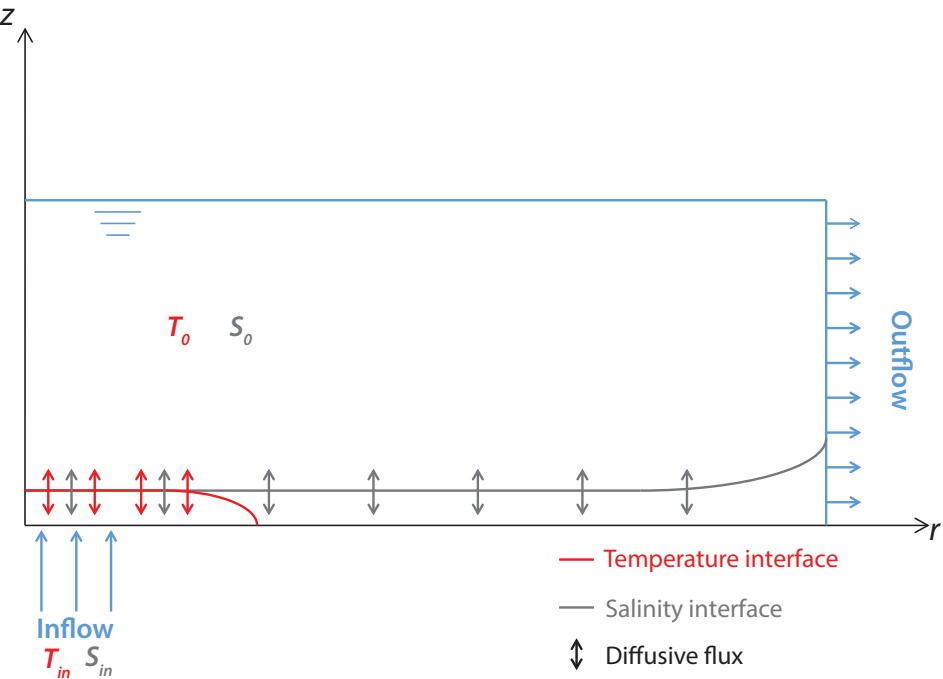

**Figure 6.** Conceptualization of the quantitative validation (Case 5), with locations of the salinity and temperature interfaces at a certain time after the start of a central inflow. The inflow is colder and more saline than the overlying water body.

(Case 5; Table 1). The interface expansion is described by its increasing interface radius $r_{int}$ over time. When the inflow is colder and more saline than the overlying water body, the developing layer has different growth rates for the salinity and temperature interface (Fig. 6). This is a consequence of the molecular heat diffusion, which is approximately 100 times larger than the diffusion of salt. In laminar flow conditions, molecular diffusion is the main driver of heat and salt exchange in stable layered systems.

In this quantitative case study, the central inflow has an outer radius of 0.2 m. To allow a slow development of the bottom layer, the inflow is placed slightly deeper compared to the rest of the bottom, and the inflow velocity linearly increases over the first 20 minutes. The discharge over the right outflow boundary is set equal to the inflow discharge:

$$Q_{out} = Q_{in} = w_{in} \cdot A_{in} \tag{32}$$

To derive the growth rates of the temperature and salinity interfaces, we consider the similarity solution of the heat equation for a fixed boundary concentration (Bergman et al., 2011):

$$c(x,t) = c_{in} + \triangle c \cdot \mathrm{erfc}\left(\frac{x}{\sqrt{4 \cdot D \cdot t}}\right) \tag{33}$$

where $x$ is the distance from the interface. $\triangle c = c_0 - c_{in}$ is the difference in concentrations (salinity or temperature) between the upper water body, represented by its initial concentration, and the inflow. The total mass $M$ that has crossed the interface is found by integration of Equation 33 over $x = 0 \to \infty$, and multiplication the growing interface surface $A_{int}$:

$$M(t) = A_{int} \cdot \int\limits_0^\infty (c - c_{in}) \, dx = A_{int} \cdot \Delta c \cdot \int\limits_0^\infty \mathrm{erfc}\left(\frac{x}{\sqrt{4 \cdot D \cdot t}}\right) dx = A_{int} \cdot \Delta c \cdot \frac{\sqrt{4 \cdot D \cdot t}}{\pi} \tag{34}$$

Derivation over time results in the time dependent mass flux over the interface:

$$\Phi_{int}(t) = \frac{dM}{dt} = \triangle c \cdot \sqrt{\frac{D \cdot t}{\pi}} \cdot \left(2 \cdot \frac{dA_{int}}{dt} + \frac{A_{int}}{t}\right) \tag{35}$$

With $A_{int} = \pi r_{int}^2$, and assuming that the interface surface increases linearly with time at a constant inflow, we can rewrite:

$$r_{int}(t) = \sqrt{\frac{\Phi_{int}}{3 \cdot \triangle c} \cdot \sqrt{\frac{t}{D \cdot \pi}}} \tag{36}$$

We assume that no mass is stored in the lower layer. Consequently, the mass flux that crosses the interface is equal to the net mass flux into the domain $\Phi_{in} - \Phi_{out} \approx w_{in} \cdot A_{in} \cdot (c_{in} - c_0)$:

$$r_{int}(t) = \sqrt{\frac{w_{in} \cdot A_{in}}{3} \cdot \sqrt{\frac{t}{D \cdot \pi}}} \tag{37}$$

This equation can be used to validate the interface growth of both the salinity and temperature interface in the case of laminar flow.

### 2.4.3 Validation for double-diffusive characteristics

The Cases 6 and 7 represent seepage inflows similar to the ones for which this modelling approach is developed. A dual-layered system is built up by a central inflow through the bottom with an outer radius of 0.25 m (Table 1). The inflow velocity $w_{in}$ is built up linearly over the first 10 minutes to prevent a sudden pressure wave at $t = 0$. Like Case 5, the average water level is kept constant by a uniform outflow with the same discharge over the right, outer boundary. Based on the Turner angle, a system with double-diffusive convection is expected to build up in Case 6, whereas a gravitationally unstable system is expected to develop in Case 7.

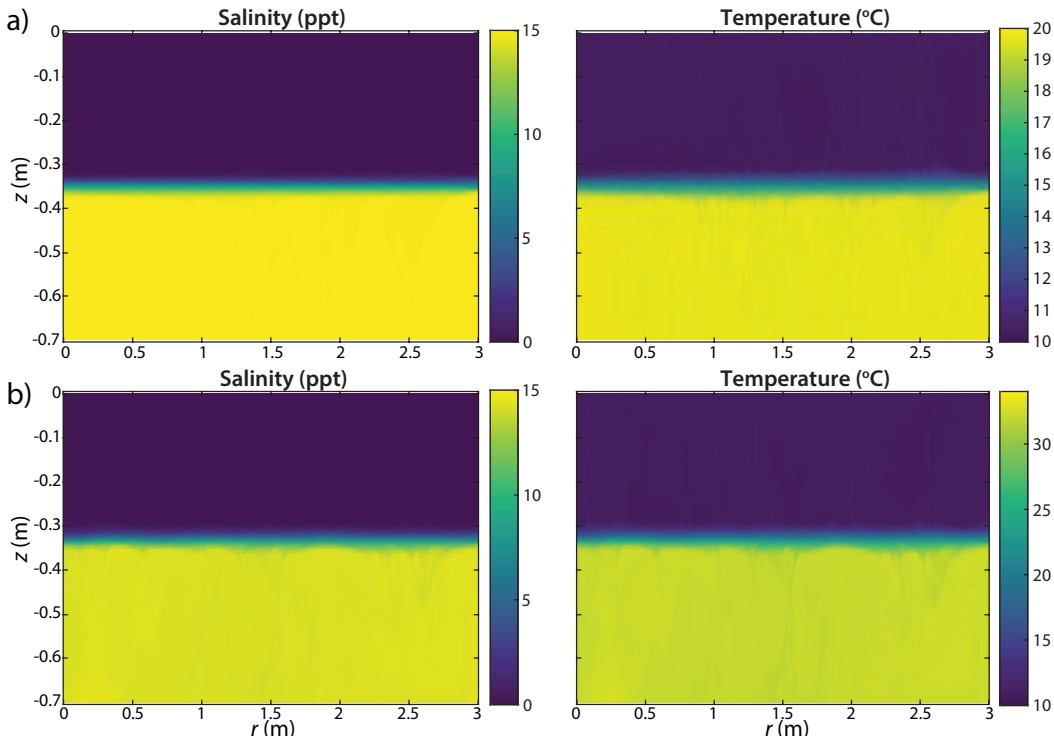

**Figure 7.** Double-diffusive convection in a layered system with a cold and fresh layer on top of a warm and saline water ($t = 4500\,$s since the start), with density ratios of a) $R_\rho = 0.15$ (Case 1), and b) $R_\rho = 0.50$ (Case 2). All figures represent simulations without the use of a turbulence model.

## 3   Results and discussion

The performance of the numerical framework was tested in several case studies subject to double-diffusive processes. The numerical results of these case studies and the extended SWASH code are presented in Hilgersom et al. (2017).

### 3.1   Case 1 and 2: Double-diffusive convection

The temperature and salinity gradients in the Cases 1 and 2 yield a theoretical onset of double-diffusive convection, with respective Turner angles of -53.3$^{\rm o}$ and -71.6$^{\rm o}$. The numerical results confirm that a layered system is maintained, bordered by a thin boundary layer from which unstable plumes emerge (Fig. 7). These are clear characteristics of double-diffusive convection.

The boundary layer thickness ratio $r$ is expected to be $\sim$2.5. Fig. 8 shows that none of the simulations for Case 1 and 2 reach this value of $r$. For Case 1, the simulation without the aid of a turbulence model reaches the highest value of $r$, although this ratio starts to decrease again 3.5 h after the start. The fact that the expected values of $r$ are not reached during the simulations seems in line, though, with Carpenter et al. (2012), who presented the evolution of $r$ in 3-D DNS simulations of a salt-heat

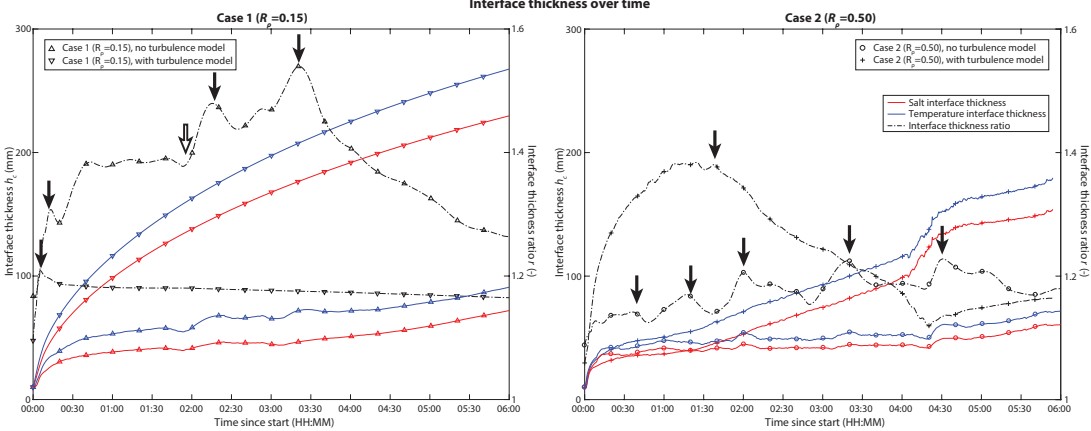

**Figure 8.** Time evolution of the interface thicknesses $r$ for the depth profiles of $S$ (red) and $T$ (blue) and their ratios, for $R_\rho = 0.15$ (Case 1, left) and $R_\rho = 0.50$ (Case 2, right). The interface thicknesses are averaged over the complete horizontal domain and each subplot presents the results for the simulations with and without a turbulence model. The filled arrows mark clear drops of $r$, and the open arrow marks the moment that $r$ starts its further increase in the direction of the theoretical interface ratio $r \approx 2.5$ in the simulation of Case 1 without a turbulence model.

system. They found that the salinity field in their simulation was not well resolved over the first 14 h. Only after this first period of high turbulence, the boundary layer thickness ratio approaches and remains its expected value.

Regarding the tendency to reach a steady state step by step by building up a system with a stable boundary layer, our findings for Case 1 with a turbulence model seem more alarming. Here, $r$ falls

back to a value of 1.18 after its first increase and remains this value afterwards. The fact that the simulation does not tend to a system with a higher value of $r$ afterwards, might indicate that the standard k-$\epsilon$ model does not function for systems with high density gradients.

The simulations for Cases 2 also show an initial increase of $r$, followed by one or more drops. The density ratio $R_\rho = 0.50$ indicates a mere turbulent system. In this sense, it is expected that

the boundary layers do not develop easily. Here, the simulation with a turbulence model seems to develop $r$ as expected over the first 1.5 h. However, the ratio drops to values below $r = 1.2$ afterwards.

The poor performance of the standard k-$\epsilon$ model also appears from the exaggerated salt and heat fluxes for Case 1 (Figure 9b) compared to the simulation without the turbulence model (Figure 9a).

In the latter simulation, the heat transport over the interface appears to follow the theoretical heat transport as predicted from the molecular heat diffusion. Based on the flux ratio equations by Kelley (1990) and Fernando (1989) (Eq. 25 and Eq. 26), we expect that the salt transport across the interface is lower than the heat transport. Initially, this is not the case (Figure 9a), but the salt flux approaches the values predicted by Fernando (1989) after 110 (min). This moment coincides with the moment

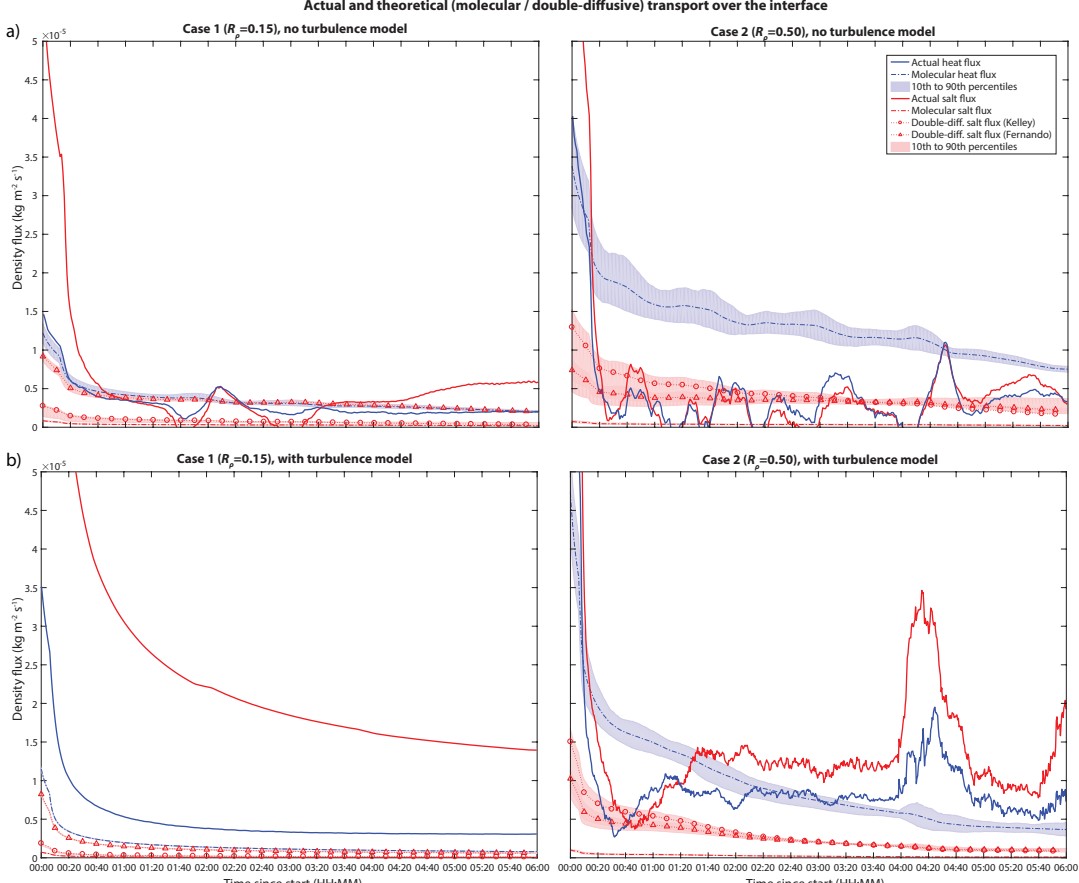

**Figure 9.** Time evolution of the salt (red) and temperature fluxes (blue) over the interface for $R_\rho = 0.15$ (Case 1, left) and $R_\rho = 0.50$ (Case 2, right), and for a) the simulations without a turbulence model and b) the simulations with a turbulence model. The fluxes represent horizontal averages throughout the complete domain. The figures also present theoretical fluxes over the interface, which were calculated from molecular and double-diffusive diffusivities, as well as their uncertainty bounds for horizontal variations in density gradients at the interface.

that $r$ starts a sharp increase in Fig. 8. In general, the salt flux has expected lower values than the heat flux over the period that the ratio $r$ is highest.

The ratio of the simulated heat fluxes to the 4/3 flux law (Eq. 27 to 29), shows a similar jump after 110 min (Fig. 10). From this moment, the simulated heat flux in Case 1 without a turbulence model temporarily approaches the predicted heat flux of Kelley (1990) and Linden and Shirtcliffe (1978). Again, a similar tendency is not visible for the simulation with a turbulence model, confirming that the standard k-$\epsilon$ model suppresses the onset of double-diffusive convection.

In line with the expectations for turbulent flows, the simulations for Case 2 show a large variation in heat and salt transport (Fig. 9). The simulations with and without a turbulence model both display a heat flux that is variably higher and lower than the salt flux, but displays the same pattern. Theoret-

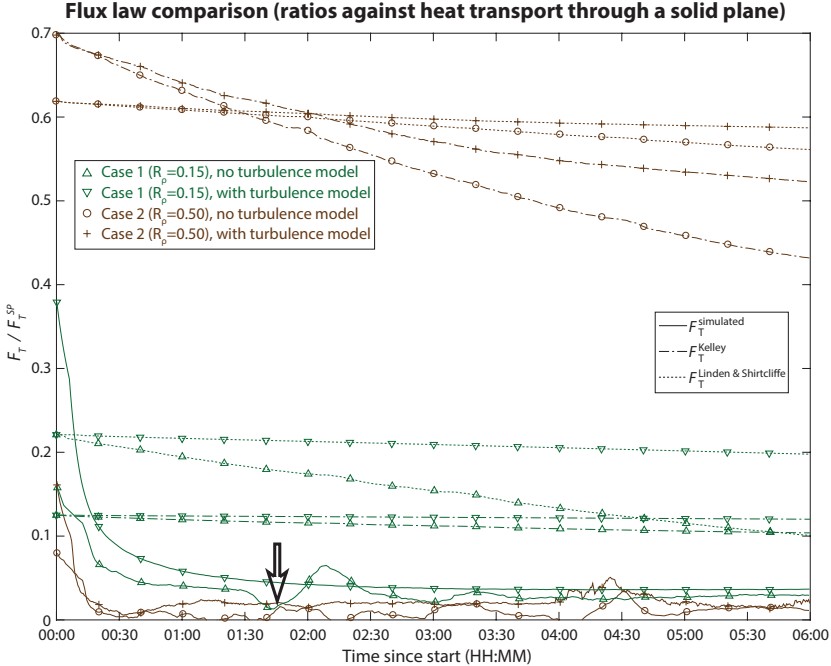

**Figure 10.** Simulated and theoretical heat fluxes according to Eq. 27 and Eq. 28, relative to the theoretical heat flux through a solid plane (Eq. 29). Green colours represent the results for Case 1 and brown colours represent the results for Case 2. The open arrow marks the moment that the simulated heat flux starts rising towards the theoretical flux laws in the simulation of Case 1 without a turbulence model.

ically, the ratio of the turbulent heat and salt fluxes across the boundary approaches $\tau^{\frac{1}{2}}$ (Fernando, 1989) as $R_\rho$ approaches unity.

The dissimilar behaviour of our simulation with a turbulence model can be explained by the employed eddy diffusivities which have similar values for salt and heat diffusion (note that the turbulent Prandtl and Schmidt number have similar values). These eddy diffusivities were not employed in the

simulation without a turbulence model, which indicates that the similar heat and salt transport across the interface is caused by turbulent mixing through this interface. We refer to Section 3.6 for a further discussion on this in light of the employed standard k-$\epsilon$ model.

### 3.2 Case 3 and 4: Salt-fingers

The numerical results for Case 3 ($Tu = 71.2°$) and Case 4 ($Tu = 85.0°$) confirm that salt-fingers

are formed over the interface (Fig. 11). Based on the difference in density ratios, the salt-fingers in Case 4 are expected to transport more salt and heat than Case 3 (Section 2.4). Fig. 5 shows an interface rise of about $0.04$ m in Case 3 and $0.13$ m in Case 4 over a numerical model run of 6 h. Given the system of closed boundaries, we therefore find a significantly larger transport over the interface in Case 4.

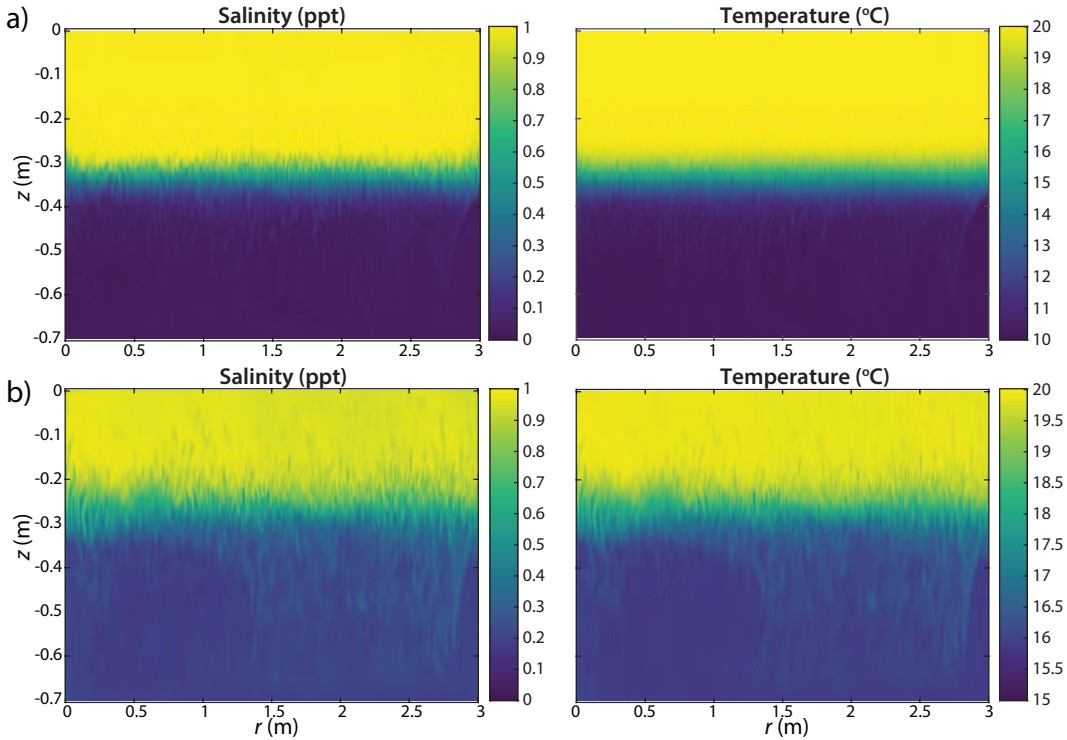

**Figure 11.** Salt-fingering in a layered system with a warm and saline water on top of a cold and fresh layer ($t$ = 4500 s since the start), with density ratios of a) $R_\rho = 2.04$ (Case 3), and b) $R_\rho = 1.19$ (Case 4). All figures represent simulations without the use of a turbulence model.

Similar to the Cases 1 and 2 (Fig. 9), we calculated the salt and heat fluxes across the interface for Case 3 and Case 4. For these cases, however, we only report the simulated flux ratios $\gamma_{sim}$ of the heat and salt fluxes (Fig. 12). For salt-fingers, flux ratios lower than unity are expected from Eq. 24. From our simulations, however, we find flux ratios higher than unity. These flux ratios are more in line with oceanic values where turbulent values of $\gamma$ can approach 1.6 (Kunze, 2003). Over

the simulated 6 hours, the flux ratios show a decreasing tendency. However, particularly for the mere turbulent Case 3, we observe sudden upward jumps in the flux ratios, preventing the flux ratios to reach consistent values below unity over the course of the simulations. Based on these results, we hypothesize a settling of the system with more constant low flux ratios on the long run.

     Based on a 3-D DNS model, Traxler et al. (2011) found Stern numbers $St = 9.4$ and $St = 76$

for $R_\rho = 2.0$ and $R_\rho = 1.2$, respectively. Our simulations for $R_\rho = 2.04$ (Case 3) and $R_\rho = 1.19$ (Case 4) yield lower Stern numbers: on average approximately 0.73 and 1.95, respectively (Fig. 13). One reason for the lower values could be found in the fact that our model simulates salt and heat transport in two dimensions: Traxler et al. (2011) reported a Stern number of approximately 3.5 in 2-DV simulations for $R_\rho = 2.0$. Although our simulations yield even lower Stern numbers, values

around $St = 1$ are not uncommon for salt-finger systems.

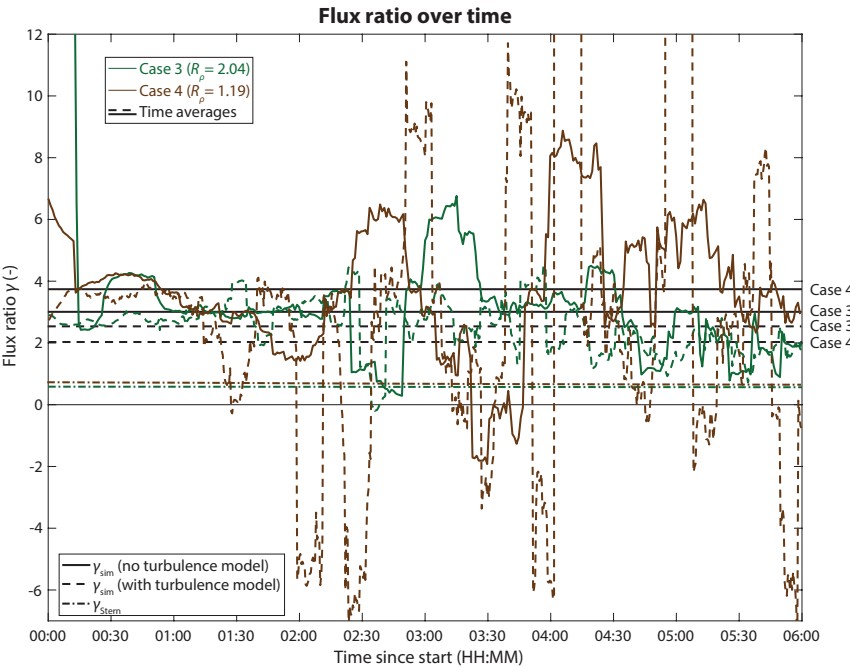

**Figure 12.** The evolution of flux ratios over time for $R_\rho = 2.04$ (Case 3, green) and $R_\rho = 1.19$ (Case 4, brown). Continuous lines mark the simulations without a turbulence model, and dotted lines marks the simulations with a turbulence model. The black lines mark the temporal averages for each simulation.

### 3.3 Case 5: Radial expansion of a dense water layer

The analytical solution for the radial expansion of inflowing cold and saline water (Equation 37) holds for a situation with laminar flow. Given the geometric properties of the conceptualized situation and the initially very thin layer of dense water, it is difficult to define the inflow properties so that the flow near the inflow is immediately laminar. For the selected inflow parameters (Table 1), laminarisation of the flow appears to occur after approximately 1700 s (Fig. 14). From that moment, the numerical results show significant differences between the salinity and temperature interface growth. The analytical results are therefore shifted in time to match the interface radii with the numerical results at the moment that the flow becomes laminar.

Accounting for a purely molecular diffusion, the numerical results show a fair agreement with the analytical results. As we found some small occasional eddies occurring after $t = 1700$ s, we also plotted results analytical results assuming the diffusivity was on average for 0.2 % influenced by turbulent diffusion. Here, the turbulent diffusion was calculated by dividing an assumed kinematic viscosity $\nu = 10^{-6} \text{ m}^2\text{s}^{-1}$ by the Prandtl-Schmidt number (Equations 8 and 9. The assumption of a slight influence of turbulence diffusion shows a better agreement with the numerical results.

One critical note here is the sensitivity of the interface growth to the definition of the interface location. Similar to the previous cases, we defined the interface location halfway the step change

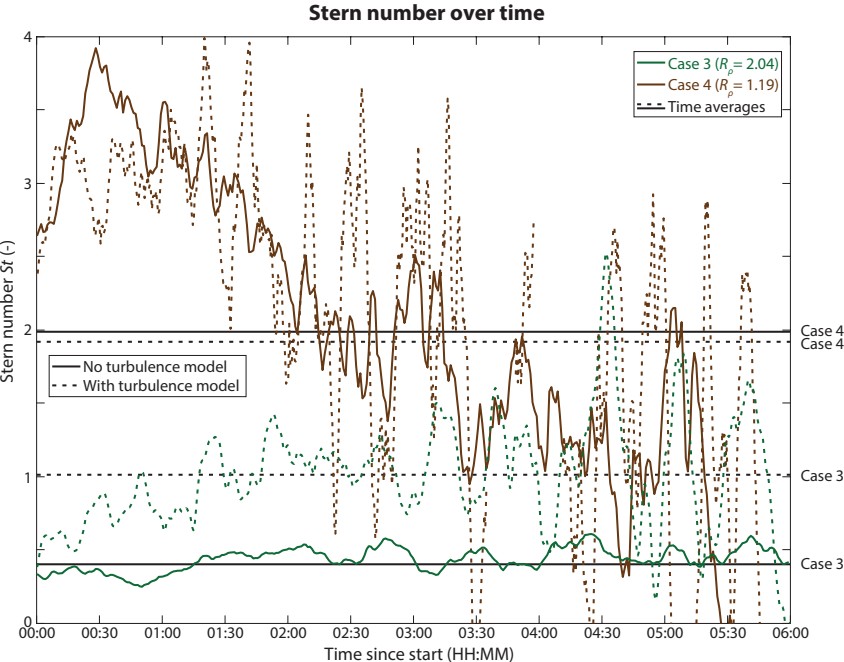

**Figure 13.** The evolution of the Stern numbers over time for $R_\rho = 2.04$ (Case 3, green) and $R_\rho = 1.19$ (Case 4, brown). Continuous lines mark the simulations without a turbulence model, and dotted lines marks the simulations with a turbulence model. The black lines mark the temporal averages for each simulation.

between the inflow concentration ($T_{in}$ and $S_{in}$) and the concentration of the water body ($T_0$ and $S_0$), because this matches our visual interpretation of the interface in the numerical results. However,
selecting the interface at a larger percentage of the step change significantly increases the growth, and makes the numerical and analytical results incomparable.

### 3.4   Case 6: Inflow yielding double-diffusive convection

The temperature and salinity gradients in Case 6 yield the onset of double-diffusive convection. Like the Cases 1 to 4, a sharp interface develops over which salt and heat is transported by diffusion.
Fig. 15 confirms the development of a salt-heat interface and a convective layer above the boil. Other convective cells further transport the salt and heat above the interface. Fig. 15 shows that already a considerable amount of heat and salt was conveyed to the upper layer over the first 1.5 h. The lower convective layer slowly builds up, and local eddies clearly counteract the development when the lower convective layer is still thin.

### 3.5   Case 7: Gravitationally unstable inflow

Compared to Case 6, a slightly altered inflow temperature and salinity in Case 7 theoretically makes the developing layer gravitationally unstable (Table 1). In other words, the water body itself is denser

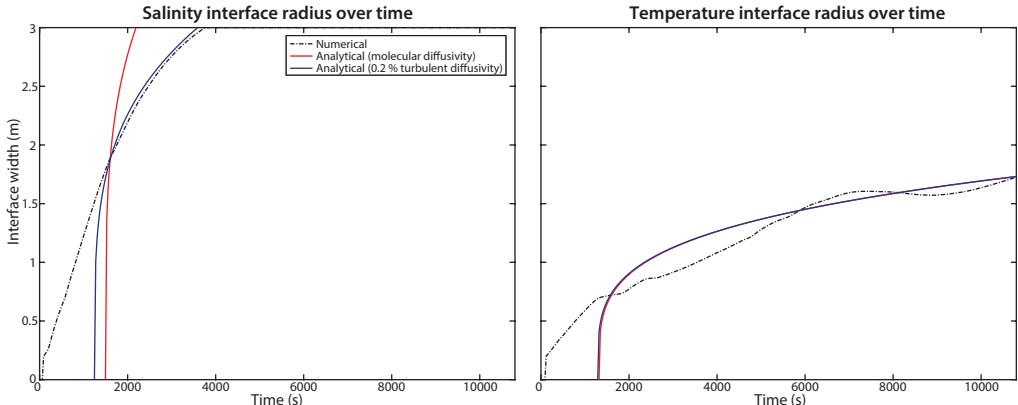

**Figure 14.** Evolution of the interface between a warm and fresh water body and a bottom cold and saline layer developing form a central inflow (Case 5). After $t = 1700$ s, the flow in the numerical results becomes laminar and differences between the temperature and salinity interface growth become visible. Analytical results are plotted for the assumptions of completely molecular diffusion (red), and for diffusivities that are for 0.2 % influenced by turbulent diffusion (blue).

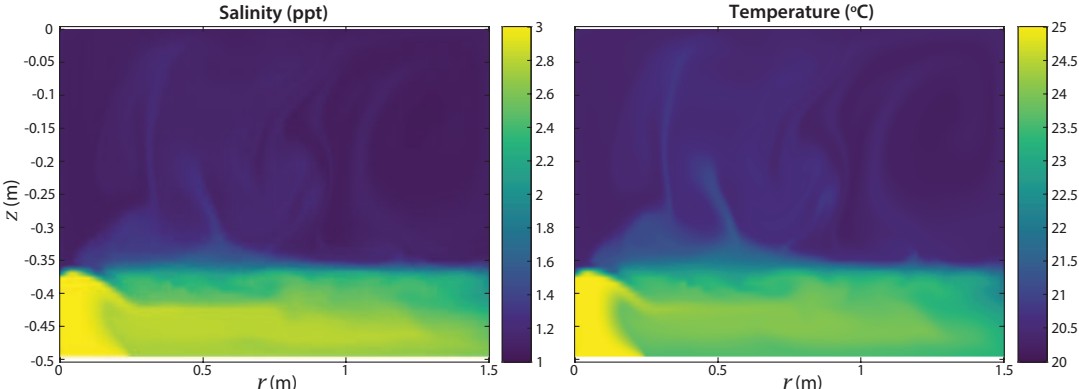

**Figure 15.** Double-diffusive layering (Case 6) with cold and fresh water on top of a warm and saline inflow ($t = 5400$ s since the start).

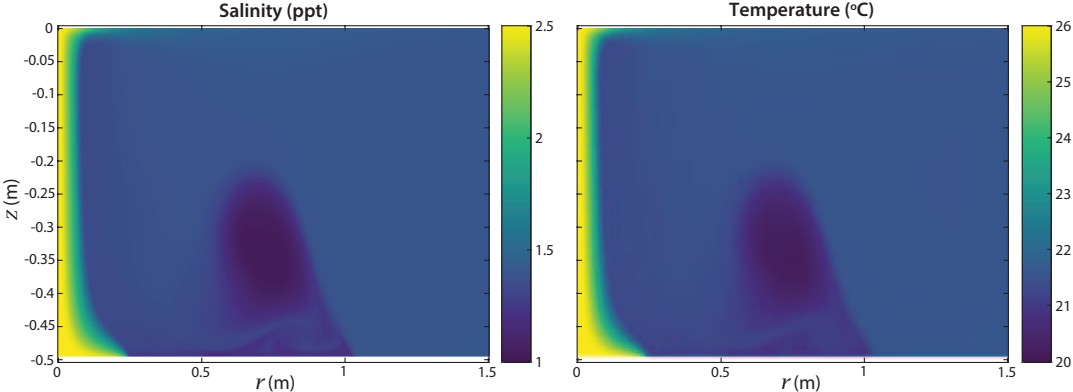

**Figure 16.** Unstable system (Case 7) with denser cold and fresh water on top of a warm and saline inflow ($t = 5400$ s since the start). The inflowing water flows upward through the centre, independent of the inflow velocity.

than the inflowing water, which consequently flows upwards. The numerical results confirm the onset of a central buoyant flow above the inflow (Fig. 16).

Interestingly, plumes develop from the upward flow. Downward plumes are also visible below the floating warm and saline water. Like the salt-fingers in Case 3 and 4, where warm and saline water also overlaid cold and fresh water, this is a mechanism to dissipate the heat and salt gradients.

### 3.6    Turbulence model

In the previous subsections, we found that the standard k-$\epsilon$ model performed insufficiently accurate
for predicting production and dissipation of turbulence. This subsection briefly discusses the performances of the turbulence model and future prospects for a relatively simple improvement of the model.

In Section 3.1, we found a similar heat and salt transport across the interface that was likely caused by turbulent mixing at the interface. Following this hypothesis, the RANS model apparently
requires a turbulence model that suppresses the turbulence across the interface, but predicts the onset of turbulence in the unstable regions near the interface. This is a known defect of the standard k-$\epsilon$ model, which does not account for buoyancy effects near strong density gradients. Section 3.3 also stressed the importance of a right timing on which turbulence is modelled when flows are variably laminar and turbulent.

More advanced turbulence models have been developed for systems with large density gradients (e.g., Venayagamoorthy, 2003; Paik et al., 2009). Toffolon et al. (2015) recently showed how a minimal model with two parameters is already able to characterize differences in transport between sharp interfaces and unstable regions in a thermohaline staircase. Extending the k-$\epsilon$ model of SWASH with a parametrization that accounts for these distinct regions could yield a large improvement in properly

representing turbulence on proper locations and times. With such an advanced turbulence model, the largest improvements are expected when $R_\rho$ approaches unity, as a good turbulence model becomes increasingly important for these density gradients.

## 4   Conclusions

This article reports the successful derivation of an axisymmetric framework for a hydrodynamic
model incorporating salt and heat transport. This model set-up allows to efficiently calculate salt and heat transport whenever a situation is modelled that can be approximated by axisymmetry around a central location. The 2-D axisymmetric grid description demands approximately the same execution time as a regular 2-DV description with the same dense mesh, and therefore avoids the need to solve the equations over a dense mesh in the third spatial dimension.

For our purpose of studying shallow water bodies, three aspects were important: 1) the inclusion of a free surface, 2) the efficient solution of a circular seepage inflow, which makes the problem three-dimensional, and 3) a proper simulation of density driven flow and double-diffusivity driven salt and heat transport. The former aspect was already fulfilled by employing the SWASH framework.

    The second aspect was solved by assuming axisymmetry for the Reynolds-averaged Navier-Stokes
equation in cylindrical coordinates. The derived numerical framework is presented as a Cartesian 2-DV description with few additional terms and width compensation factors. Our implementation of these terms in the non-hydrostatic SWASH model demonstrates the opportunity to easily extend a 2-DV model towards the presented 2-D axisymmetric model.

    The third aspect was fulfilled by extending SWASH with a new density and diffusivity module.
The case studies demonstrate explainable behaviour for density driven flow and double-diffusivity driven salt and heat transport. The formation of convective layers and salt-fingers themselves are in accordance with the theory of double-diffusivity, as well as the enhanced salt and heat fluxes across the interface for density gradients approaching unity. Other validation metrics show that the RANS model does not meet the expected flux ratios and stability criteria in all cases, which is hypothesized
to be caused by a defective turbulence modelling for systems of large density gradients. Replacing the standard k-$\epsilon$ model by an advanced turbulence model might improve the results for these merely turbulent cases.

    An analytic validation method was presented to evaluate the model's performance for a cold and saline inflow developing a dense water layer near the bottom. For laminar flow conditions, the nu-
merical model showed a similar radial expansion of the bottom layer as expected from analytical results.

    Although the model is already able to show expected behaviour in the double-diffusive regime, we recommend a further exploration of its limitations and possibilities. For example, a grid convergence study should indicate whether the selected mesh size yields a convergence of results for all diffusion

and advection dominated cases. Further, a nearer comparison with DNS model results would support the validation of the model. In future applications, we stress that this model approach should be employed as a RANS model that simulates thermohaline stratification processes on a larger scale. As such, the model can be favourable in applications that allow an axisymmetric approach.

**Data availability**

The model data for the five case studies and the extended SWASH code are accessible on doi:10.4121/uuid:95227d5d-2cf0-44ec-ab2d-705a626dcdf4 (Hilgersom et al., 2017).

**Appendix A: Cell depth integration with the Leibniz integral rule**

When the continuity, momentum and transport equations are integrated over the cell depth, the Leibniz integral rule is applied to the time derivatives and the horizontal spatial derivatives. Here, we

show the cell depth integration of $\frac{\partial u}{\partial t}$ and $\frac{\partial uu}{\partial r}$:

$$\int_{z_{k-\frac{1}{2}}}^{z_{k+\frac{1}{2}}} \frac{\partial u}{\partial t} dz = \frac{\partial u_k h_k}{\partial t} - u \frac{\partial z}{\partial t}\bigg|_{z_{k-\frac{1}{2}}}^{z_{k+\frac{1}{2}}} \tag{A1}$$

$$\int_{z_{k-\frac{1}{2}}}^{z_{k+\frac{1}{2}}} \frac{\partial uu}{\partial r} dz = \frac{\partial u_k u_k h_k}{\partial r} - u\hat{u} \frac{\partial z}{\partial r}\bigg|_{z_{k-\frac{1}{2}}}^{z_{k+\frac{1}{2}}} \tag{A2}$$

The derivatives $\frac{\partial ur}{\partial r}$, $\frac{\partial w}{\partial t}$, $\frac{\partial uw}{\partial r}$, $\frac{\partial p}{\partial r}$, $\frac{\partial c}{\partial t}$, and $\frac{\partial c}{\partial r}$ in Equations 1, 2, 3, and 7 are integrated in a similar fashion.

## 545 Appendix B: Full discretizations

### B1 U-momemtum

$$\frac{u_{i+\frac{1}{2},k}^{n+\theta_u}}{\theta_u \Delta t} + \frac{\overline{\omega_{i+\frac{1}{2},k+\frac{1}{2}}^{n}}^r \left(\widehat{u}_{i+\frac{1}{2},k+\frac{1}{2}}^{n+\theta_u} - u_{i+\frac{1}{2},k}^{n+\theta_u}\right)}{\overline{h_{i+\frac{1}{2},k}^{n}}^r} - \frac{\overline{\omega_{i+\frac{1}{2},k-\frac{1}{2}}^{n}}^r \left(\widehat{u}_{i+\frac{1}{2},k-\frac{1}{2}}^{n+\theta_u} - u_{i+\frac{1}{2},k}^{n+\theta_u}\right)}{\overline{h_{i+\frac{1}{2},k}^{n}}^r} -$$

$$\nu_{v;i+\frac{1}{2},k+\frac{1}{2}}^{n} \frac{u_{i+\frac{1}{2},k+1}^{n+\theta_u} - u_{i+\frac{1}{2},k}^{n+\theta_u}}{\overline{h_{i+\frac{1}{2},k}^{n}}^r \overline{h_{i+\frac{1}{2},k+\frac{1}{2}}^{n}}^{rz}} + \nu_{v;i+\frac{1}{2},k-\frac{1}{2}}^{n} \frac{u_{i+\frac{1}{2},k}^{n+\theta_u} - u_{i+\frac{1}{2},k-1}^{n+\theta_u}}{\overline{h_{i+\frac{1}{2},k}^{n}}^r \overline{h_{i+\frac{1}{2},k-\frac{1}{2}}^{n}}^{rz}} = \frac{u_{i+\frac{1}{2},k}^{n}}{\theta_u \Delta t} -$$

$$\frac{\overline{\phi_{i+1,k}^{n} \overrightarrow{y_{1;i+1}}}^r \left(\widehat{u}_{i+1,k}^{n} - u_{i+\frac{1}{2},k}^{n}\right) - \overline{\phi_{i,k}^{n} \overrightarrow{y_{1;i}}}^r \left(\widehat{u}_{i,k}^{n} - u_{i+\frac{1}{2},k}^{n}\right)}{\overline{h_{i+\frac{1}{2},k}^{n}}^r \overline{y_{1;i+\frac{1}{2}}}^r \Delta r_{i+\frac{1}{2}}} + \boxed{\frac{\alpha}{\overline{y_{1;i+\frac{1}{2}}}^r} u_{i+\frac{1}{2},k}^{n} u_{i+\frac{1}{2},k}^{n}} -$$

$$g\frac{\zeta_{i+1}^{n+\theta_\zeta} - \zeta_i^{n+\theta_\zeta}}{\Delta r_{i+\frac{1}{2}}} - \frac{q_{i+1,k}^{n} h_{i+1,k}^{n} y_{1;i+1} - q_{i,k}^{n} h_{i,k}^{n} y_{1;i}}{\overline{h_{i+\frac{1}{2},k}^{n}}^r \overline{y_{1;i+\frac{1}{2}}}^r \Delta r_{i+\frac{1}{2}}} + \overline{q_{i+\frac{1}{2},k+\frac{1}{2}}^{n}}^{rz} \frac{z_{i+1,k+\frac{1}{2}}^{n} - z_{i,k+\frac{1}{2}}^{n}}{\overline{h_{i+\frac{1}{2},k}^{n}}^r \Delta r_{i+\frac{1}{2}}} -$$

$$\overline{q_{i+\frac{1}{2},k-\frac{1}{2}}^{n}}^{rz} \frac{z_{i+1,k-\frac{1}{2}}^{n} - z_{i,k-\frac{1}{2}}^{n}}{\overline{h_{i+\frac{1}{2},k}^{n}}^r \Delta r_{i+\frac{1}{2}}} + \boxed{\frac{\alpha \overline{q_{i+\frac{1}{2},k}^{n}}^r}{\overline{y_{1;i+\frac{1}{2}}}^r}} - \frac{g}{\rho_0} \frac{\overline{h_{i+\frac{1}{2},k}^{n}}^r}{2} \frac{\rho_{i+1,k}^{n} - \rho_{i,k}^{n}}{\Delta r_{i+\frac{1}{2}}} -$$

$$\frac{g}{\rho_0} \sum_{j=1}^{k-1} \left(\overline{h_{i+\frac{1}{2},j}^{n}}^r \frac{\rho_{i+1,j}^{n} - \rho_{i,j}^{n}}{\Delta r_{i+\frac{1}{2}}} + \left(\overline{\rho_{i+\frac{1}{2},j}^{n}}^r - \overline{\rho_{i+\frac{1}{2},k}^{n}}^r\right) \frac{h_{i+1,j}^{n} - h_{i,j}^{n}}{\Delta r_{i+\frac{1}{2}}}\right) +$$

$$\frac{\nu_{h;i+1,k}^{n} y_{1;i+1} h_{i+1,k}^{n}}{\overline{h_{i+\frac{1}{2},k}^{n}}^r \overline{y_{1;i+\frac{1}{2}}}^r \Delta r_{i+\frac{1}{2}}} \frac{u_{i+\frac{3}{2},k}^{n} - u_{i+\frac{1}{2},k}^{n}}{\Delta r_{i+1}} -$$

$$\frac{\nu_{h;i,k}^{n} y_{1;i} h_{i,k}^{n}}{\overline{h_{i+\frac{1}{2},k}^{n}}^r \overline{y_{1;i+\frac{1}{2}}}^r \Delta r_{i+\frac{1}{2}}} \frac{u_{i+\frac{1}{2},k}^{n} - u_{i-\frac{1}{2},k}^{n}}{\Delta r_i} - \boxed{\frac{\alpha}{\overline{y_{1;i+\frac{1}{2}}}^r} u_{i+\frac{1}{2},k}^{n} \frac{\overline{\nu_{h;i+\frac{1}{2},k}^{n}}^r}{r_{i+\frac{1}{2}}}} \quad \text{(B1)}$$

### B2 W-momemtum

$$\frac{w_{i,k+\frac{1}{2}}^{n+\theta_w}}{\theta_w \Delta t} + \frac{\overline{\omega_{i,k+1}^{n}}^z \left(\hat{w}_{i,k+1}^{n+\theta_w} - w_{i,k+\frac{1}{2}}^{n+\theta_w}\right)}{\overline{h_{i,k+\frac{1}{2}}^{n}}^z} - \frac{\overline{\omega_{i,k}^{n}}^z \left(\hat{w}_{i,k}^{n+\theta_w} - w_{i,k+\frac{1}{2}}^{n+\theta_w}\right)}{\overline{h_{i,k+\frac{1}{2}}^{n}}^z}$$

$$-\nu_{v;i,k+1}^{n} \frac{w_{i,k+\frac{3}{2}}^{n+\theta_w} - w_{i,k+\frac{1}{2}}^{n+\theta_w}}{\overline{h_{i,k+\frac{1}{2}}^{n}}^z h_{i,k+1}^{n}} + \nu_{v;i,k}^{n} \frac{w_{i,k+\frac{1}{2}}^{n+\theta_w} - w_{i,k-\frac{1}{2}}^{n+\theta_w}}{\overline{h_{i,k+\frac{1}{2}}^{n}}^z h_{i,k}^{n}} = \frac{w_{i,k+\frac{1}{2}}^{n}}{\theta_w \Delta t}$$

$$-\frac{\overline{\phi_{i+\frac{1}{2},k+\frac{1}{2}}^{n}}^z \overrightarrow{y_{1;i+\frac{1}{2}}} \left(\hat{w}_{i+\frac{1}{2},k+\frac{1}{2}}^{n} - w_{i,k+\frac{1}{2}}^{n}\right) - \overline{\phi_{i-\frac{1}{2},k+\frac{1}{2}}^{n}}^z \overrightarrow{y_{1;i-\frac{1}{2}}} \left(\hat{w}_{i-\frac{1}{2},k+\frac{1}{2}}^{n} - w_{i,k+\frac{1}{2}}^{n}\right)}{\overline{h_{i,k+\frac{1}{2}}^{n}}^z y_{1;i} \Delta r_i}$$

$$+\boxed{\frac{\alpha}{y_{1;i}} \frac{\overline{\phi_{i,k+\frac{1}{2}}^{n}}^{rz}}{\overline{h_{i,k+\frac{1}{2}}^{n}}^z} w_{i,k+\frac{1}{2}}^{n}} - \frac{q_{i,k+1}^{n}}{\overline{h_{i,k+\frac{1}{2}}^{n}}^z} + \frac{q_{i,k}^{n}}{\overline{h_{i,k+\frac{1}{2}}^{n}}^z} + \frac{\nu_{h;i+\frac{1}{2},k+\frac{1}{2}}^{n} \overline{y_{1;i+\frac{1}{2}}}^r \overline{h_{i-\frac{1}{2},k+\frac{1}{2}}^{n}}^{rz}}{\overline{h_{i,k+\frac{1}{2}}^{n}}^z y_{1;i} \Delta r_i} \frac{w_{i+1,k+\frac{1}{2}}^{n} - w_{i,k+\frac{1}{2}}^{n}}{\Delta r_{i+\frac{1}{2}}}$$

$$-\frac{\nu_{h;i-\frac{1}{2},k+\frac{1}{2}}^{n} \overline{y_{1;i-\frac{1}{2}}}^r \overline{h_{i-\frac{1}{2},k+\frac{1}{2}}^{n}}^{rz}}{\overline{h_{i,k+\frac{1}{2}}^{n}}^z y_{1;i} \Delta r_i} \frac{w_{i,k+\frac{1}{2}}^{n} - w_{i-1,k+\frac{1}{2}}^{n}}{\Delta r_{i-\frac{1}{2}}}$$

$$\text{(B2)}$$

 **B3 Transport equation**

$$\frac{c_{i,k}^{n+1}}{\Delta t} + \frac{\omega_{i,k+\frac{1}{2}}^{n+1}\hat{c}_{i,k+\frac{1}{2}}^{n+1}}{h_{i,k}^{n+1}} - \frac{\omega_{i,k-\frac{1}{2}}^{n+1}\hat{c}_{i,k-\frac{1}{2}}^{n+1}}{h_{i,k}^{n+1}} - \frac{D_{v;i,k+\frac{1}{2}}}{h_{i,k}^{n+1}}\frac{c_{i,k+1}^{n+1} - c_{i,k}^{n+1}}{\overline{h_{i,k+\frac{1}{2}}^{n+1}}^{z}} +$$

$$\frac{D_{v;i,k-\frac{1}{2}}}{h_{i,k}^{n+1}}\frac{c_{i,k}^{n+1} - c_{i,k-1}^{n+1}}{\overline{h_{i,k-\frac{1}{2}}^{n+1}}^{z}} = \frac{c_{i,k}^{n}h_{i,k}^{n}}{\Delta t h_{i,k}^{n}} - \frac{\phi_{i+\frac{1}{2},k}^{n}\overrightarrow{y_{1;i+\frac{1}{2}}}\hat{c}_{i+\frac{1}{2},k}^{n}}{y_{1;i}h_{i,k}^{n+1}\Delta r} + \frac{\phi_{i-\frac{1}{2},k}^{n}\overrightarrow{y_{1;i-\frac{1}{2}}}\hat{c}_{i-\frac{1}{2},k}^{n}}{y_{1;i}h_{i,k}^{n+1}\Delta r} +$$

$$\frac{D_{h;i+\frac{1}{2},k}\overline{y_{1;i+\frac{1}{2}}}^{r}\overline{h_{i+\frac{1}{2},k}^{n}}^{r}}{y_{1;i}h_{i,k}^{n+1}\Delta r}\frac{c_{i+1,k}^{n} - c_{i,k}^{n}}{\Delta r} - \frac{D_{h;i-\frac{1}{2},k}\overline{y_{1;i-\frac{1}{2}}}^{r}\overline{h_{i-\frac{1}{2},k}^{n}}^{r}}{y_{1;i}h_{i,k}^{n+1}\Delta r}\frac{c_{i,k}^{n} - c_{i-1,k}^{n}}{\Delta r} -$$

$$\frac{D_{h;i+\frac{1}{2},k}\overline{y_{1;i+\frac{1}{2}}}^{r}\overline{h_{i+\frac{1}{2},k}^{n}}^{r}}{y_{1;i}h_{i,k}^{n+1}\Delta r}\frac{\overline{c_{i+\frac{1}{2},k+1}^{n}}^{r} - \overline{c_{i+\frac{1}{2},k-1}^{n}}^{r}}{\overline{h_{i+\frac{1}{2},k+\frac{1}{2}}^{n}}^{rz} + \overline{h_{i+\frac{1}{2},k-\frac{1}{2}}^{n}}^{rz}}\frac{\overline{z_{i+1,k}^{n}}^{z} - \overline{z_{i,k}^{n}}^{z}}{\Delta r} +$$

$$\frac{D_{h;i-\frac{1}{2},k}\overline{y_{1;i-\frac{1}{2}}}^{r}\overline{h_{i-\frac{1}{2},k}^{n}}^{r}}{y_{1;i}h_{i,k}^{n+1}\Delta r}\frac{\overline{c_{i-\frac{1}{2},k+1}^{n}}^{r} - \overline{c_{i-\frac{1}{2},k-1}^{n}}^{r}}{\overline{h_{i-\frac{1}{2},k+\frac{1}{2}}^{n}}^{rz} + \overline{h_{i-\frac{1}{2},k-\frac{1}{2}}^{n}}^{rz}}\frac{\overline{z_{i,k}^{n}}^{z} - \overline{z_{i-1,k}^{n}}^{z}}{\Delta r} -$$

$$\frac{D_{h;i,k+\frac{1}{2}}}{h_{i,k}^{n+1}}\frac{z_{i+1,k+\frac{1}{2}}^{n} - z_{i-1,k+\frac{1}{2}}^{n}}{2\Delta r}\frac{\overline{c_{i+1,k+\frac{1}{2}}^{n}}^{z} - \overline{c_{i-1,k+\frac{1}{2}}^{n}}^{z}}{2\Delta r} +$$

$$\frac{D_{h;i,k+\frac{1}{2}}}{h_{i,k}^{n+1}}\left(\frac{z_{i+1,k+\frac{1}{2}}^{n} - z_{i-1,k+\frac{1}{2}}^{n}}{2\Delta r}\right)^{2}\frac{c_{i,k+1}^{n} - c_{i,k}^{n}}{\overline{h_{i,k+\frac{1}{2}}^{n}}^{z}} -$$

$$\frac{D_{h;i,k-\frac{1}{2}}}{h_{i,k}^{n+1}}\left(\frac{z_{i+1,k-\frac{1}{2}}^{n} - z_{i-1,k-\frac{1}{2}}^{n}}{2\Delta r}\right)^{2}\frac{c_{i,k}^{n} - c_{i,k-1}^{n}}{\overline{h_{i,k-\frac{1}{2}}^{n}}^{z}} \tag{B3}$$

*Acknowledgements.* This project has been funded by The Netherlands Organisation for Scientific Research (NWO), project number 842.00.004.

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
