# Peer review of "An axisymmetric non-hydrostatic model for double-diffusive water systems"

_Geoscientific Model Development, 2016_

## Referee Comment (RC1) · Anonymous Referee #1 · 13 Oct 2016

This paper develops an axisymmetric non-hydrostatic model for simulating double-diffusive processes. The governing equations, numerical schemes and test cases are clearly presented. However, I have few major concerns as follows:

1. It is not clear what the main objectives are in the paper. The test cases are 2DV. Why do you have to solve the equations in cylindrical coordinates? The reason for developing a non-hydrostatic model in cylindrical coordinates should be clearly stated.

2. To my knowledge, double diffusion is sensitive to turbulence models. Usually large-eddy simulations are conducted to capture the instability. However, no turbulence model is presented in the paper.

3. The sensitivity of the numerical results on grid should also be discussed. Since the numerical diffusion would contaminate the physics.

[Figure]

4. eq. (12): in 2DV and 3D models, bottom friction is usually accounted for through a bottom roughness. Chezy coefficient is often used in 2DH models. Why do you choose Chezy coefficient instead of bottom roughness? How does this coefficient affect your results?

5. theta is used for the tangential direction in section 2.1. However, this becomes alpha in section 2.3. Please make it consistent throughout the paper.

---

## Referee Comment (RC2) · Anonymous Referee #2 · 17 Oct 2016

The paper describes a 2-DV formulation of the equations governing double-diffusive problems. The paper is focused on mathematical details, and the physics of the investigated problem is not properly considered. In my opinion, the paper cannot be accepted for publication in the present form.

Major remarks

1) What is novel in an axisymmetric model? As pointed out also by Referee #1, the main objectives are not clear. Moreover, practical applications are not discussed (see also next comment).

2) The necessary assumption to formulate the 2-DV model is axial symmetry. However, there is no discussion whether such a symmetry exists in real double-diffusive cases. For instance, the axial symmetry implies that salt fingers are not real "fingers", but

"circles" that develop around a central location. Is this reasonable? This is probably the major limitation of this work.

3) The model is not a DNS model, but a standard RANS model with a k-epsilon model. This means that results are dependent on the parameterization of turbulence, and that the model requires calibration and validation. This is an even more demanding issue in double-diffusive phenomena, which are at the transition between laminar and weakly turbulent flows. I believe that a standard k-epsilon model is not suitable for these conditions, so the whole model formulation is questionable. At least, it cannot be sold as a model that does not require calibration.

4) No comparison is provided with laboratory and/or numerical experiments. Only qualitative analogies are discussed, apart from the case of the central inflow (which is likely dominated by advection and not double diffusion). The authors should try to validate their results at least against DNS.

5) One of the major advantages of this formulation is the consideration of the free surface. However, I cannot see where this is a crucial aspect in double-diffusive problems. To my knowledge, these phenomena occur in deep water and are typically not influenced by the dynamics of the free surface, so the authors should explain why this characteristic is important.

6) The formulation contains some errors (see comments below).

7) The literature review is incomplete and, especially for double diffusion in the diffusive regime, outdated. For instance, no reference is given to recent DNS work, both 2D (e.g., Noguchi & Niino, 2010a,b) and 3D (e.g., Kimura & Smyth, 2007; Carpenter et al., 2012, Sommer et al., 2014). Moreover, papers that analyze the thermohaline staircase (e.g., Radko et al., 2014a,b) could be used to find cases to compare with.

Detailed comments

- l. 10, what is a "diffusivity driven flow"? Double diffusion is due to differential diffusivity,

but the flow is always driven by density gradients.

- l. 57-58, "the momentum and mass conservative grid setup allows accurate modelling of transport processes": not clear.

- l. 70, "model code does not require the calibration": this would be true for a DNS, but certainly there are parameters that are not exact in the turbulence description used in this model (see also one of the major remarks).

- l. 71, "validated": I cannot see any real validation of the model in this paper.

- l. 88-90, "the horizontal kinematic viscosity nu_h is set uniform to its molecular value ($\sim 10-6$ m2s$-1$). The non-uniform vertical viscosity nu_v includes the local eddy viscosity, as calculated by the standard k-epsilon model": why the horizontal viscosity should be characterized by its molecular value and the turbulent component added only to the vertical viscosity? I cannot see any reason why local turbulence (if present in this small-scale problems) should be accounted only in one direction.

- l. 108-109, "To account for turbulent diffusion, D_h and D_v are calculated by adding the molecular diffusivities and turbulent diffusivities: D = D_mol+D_turb. The turbulent diffusivities are calculated by dividing the eddy viscosity nu_turb by the turbulent Prandtl number". Does this sentence imply that the eddy viscosities are isotropic? (this is different from the previous point)

- eq. 6 is wrong. Not only there is a typo, i.e. "dQ/dt" is "dQ/dr", but also the structure of the equation is wrong, inasmuch it is not formulated the cylindrical coordinate system.

- l. 117, "molecular heat and salt diffusion rates, which in turn are highly dependent on temperature and salinity": I agree that there is a clear dependence on temperature, but the authors should explain why the dependence is relevant in double-diffusive problems that are typically characterized by very small temperature differences. In this respect, the dependence of density on temperature should be more important.

- l. 136, "dc/dr=0": this boundary condition is wrong in cylindrical coordinates.

- eq. 13: "omega" is not defined; why "dy" and not "dr".

- l. 164, "anti-creepage terms": explain what are these terms.

- eqs. 20 and 21: units are missing.

- l. 214-215, why is the "central inflow" a representative case? (apart from being an axisymmetric case)

- fig. 5, what does the asterisk means in the caption "depth (*)"?

- l. 261-262, "the salt-fingers in Case 2 are hypothesized to transport more salt and heat": this is an example of vague statements that are common in this paper. Why "hypothesized"? Can we see some numbers?

- l. 266-269: here the authors seem to be aware that there is a problem in describing salt fingers using axial symmetry. As I already pointed out, a field of salt fingers is not axisymmetric (a single finger can be, but not a number of them). This is one of the major drawback that needs to be clarified.

- fig. 7, where is the interface located? In a single point? Is it an average?

- l. 271-272 and fig. 8: where is the layer structure typical of double-diffusive processes?

- l. 290-291, "laminarisation": I do not understand what the authors refer to. Which kind of flow exists before being laminar?

- l. 297, "diffusivity was on average for 0.5 % influenced by turbulent diffusion": this is a very strange approach. If the turbulence model is working properly (which is the implicit assumption to be confident in using it), why reducing the calculated value of turbulent diffusivity?

- l. 299, "by applying the Prandtl-Schmidt number": the procedure is not clear.

- l. 310, "whenever a situation is modelled that can be approximated by axisymmetry

around a central location": please provide evidences that this situation exists and is relevant.

- l. 320-321, "The formation of convective layers and salt-fingers are in accordance with the theory of double-diffusivity": this statement is not demonstrated.

- l. 321, "quantitative validation method": I cannot see any general method for validating the model results in this paper. If the authors refer only to the case for the central inflow and to the fact that "the numerical model showed a similar radial expansion of the bottom layer as expected from analytical results" (l. 324), the result is not enough to justify the publication of this paper, considering also that the comparison is satisfactory only for a specific definition of the interface (l. 304-306).

References

Carpenter, J.R., Sommer, T., Wüest, A. (2012), Simulations of a double-diffusive interface in the diffusive convection regime, Journal of Fluid Mechanics, 711, pp. 411-436

Kimura, S., Smyth, W. (2007), Direct numerical simulation of salt sheets and turbulence in a double-diffusive shear layer, Geophysical Research Letters, 34 (21), L21610.

Noguchi, T., Niino, H. (2010a), Multi-layered diffusive convection. Part 1. Spontaneous layer formation, Journal of Fluid Mechanics, 651, pp. 443-464.

Noguchi, T., Niino, H. (2010b), Multi-layered diffusive convection. Part 2. Dynamics of layer evolution, Journal of Fluid Mechanics, 651, pp. 465-481.

Radko, T., Bulters, A., Flanagan, J.D., Campin, J.-M. (2014a), Double-diffusive recipes. Part I: Large-scale dynamics of thermohaline staircases, Journal of Physical Oceanography, 44(5), 1269–1284. Radko, T., Flanagan, J.D., Stellmach, S., Timmermans, M.-L. (2014b), Double-diffusive recipes. Part II: Layer-merging events, Journal of Physical Oceanography, 44(5), 1285–1305.

Sommer, T., Carpenter, J.R., Wüest, A. (2014), Double-diffusive interfaces in Lake Kivu

reproduced by direct numerical simulations, Geophysical Research Letters, 41 (14), pp. 5114-5121.

**[GMDD](https://www.geosci-model-dev-discuss.net/)**

---

## Referee Comment (RC3) · Anonymous Referee #3 · 18 Oct 2016

The paper presents a new code capable of simulating nonhydrostacy due to salt and heat double diffusion, providing the numerical methods and example test cases. Existing challenges due to small time steps are circumvented by the model formulation. The paper is clearly written and model development appears to be sound, although additional support for the paper could demonstrate numerical robustness via convergence studies as well as demonstrated applicability to the real world via comparison with laboratory study data.

General comments:

* It would be good to elaborate on use of SWASH vs a completely new model. Was the primary reason to take advantage of computational infrastructure? It seems like this may have been more work than starting fresh and it would be nice to include

further details for this design choice. * Please justify this choice of method as opposed to alternative methods, e.g., advanced mesh refinement * Please provide additional discussion of applications and uses for this code.

Specific comments:

line 146: may be nice to state briefly why staggered grid is used for this case (especially the connection to nonhydrostatic pressure and avoidance of pressure modes) line 150-175: potentially include in appendix and summarize here for brevity line 181: please justify use of poor time accuracy Euler explicit method line 198: please remove dots from equations because this makes it harder to read (this is true for other places too) Figure 4: variable line type widths would make this figure clearer Colorbars throughout: please do not use jet. Please use an alternative color bar that is not artificially misleading, e.g., gray or viridis. * Demonstration of model convergence would be very helpful in providing confidence in results * Please discuss in more detail the simpliciations resulting from the axisymmetric assumption used in this paper because it is not too hard to believe that the problem may not be axissymtetric. Additional justification will clarify the paper, e.g., in the results section. * Conclusions would benefit by returning to the question of how this model could be used to better understand double diffusion or real world problems.

Technical corrections:

line 64: axisymmetric equation 6: please place on two separate lines for clarity line 140: in the tangential direction

---

## Author Comment (AC1) · 7 Dec 2016

First of all, we would like to thank Referee #1 for reading the manuscript carefully and expressing his/her thoughts on where the manuscript should be improved. We hope that our answers and improved submission take away most of the referee's major concerns. In the following, we answer the comments point by point (Section 1) and provide an overview of the changes made to the manuscript based on these comments (Section 2).

[Figure]

**1 Comments and answers**

*1. It is not clear what the main objectives are in the paper. The test cases are 2DV. Why do you have to solve the equations in cylindrical coordinates? The reason for developing a non-hydrostatic model in cylindrical coordinates should be clearly stated.*

This article explores the density-driven flow in radial direction around a central seepage source. In two of the test cases (Case 1 and 2), this inflow is absent. However, these test cases also serve to test the functioning of the axisymmetric model set-up.

The introduction of the article already explained why an hydrodynamic model in cylindrical coordinates can be preferential in some specific cases (lines 50-53). We found reason to develop a non-hydrostatic model in cylindrical coordinates, because these axisymmetric cases exist, for example in river deltas with saline seepage. The presented model set-up allows to correctly represent the volumetric (in)flow in the model. In the updated version of our article, we therefore extended the paragraph (lines 50-53) to clearly state the existence of axisymmetric cases. The following text will replace the last sentence of this paragraph: "Examples of such cases are close-to-circular water bodies with uniform boundaries, and the flow around a central point (e.g., a local inflow from a pipe or groundwater seepage). The occurrence of local saline seepage inflows into shallow water bodies of contrasting temperatures has been described by De Louw et al. (2013). Hilgersom et al. (2016) have shown how these local inflows can induce thermohaline stratification in the shallow surface water bodies above these inflows." The requirement to correctly represent the volumetric flow in the modelling approach will now be better stated in lines 63-64, where we explain why we develop an axisymmetric variation of SWASH (see our answer to Referee #2).

*2. To my knowledge, double diffusion is sensitive to turbulence models. Usually largeeddy simulations are conducted to capture the instability. However, no turbulence model is presented in the paper.*

We agree that the inclusion of turbulence is important when modelling double diffusion, and therefore our simulations did employ a turbulence model. In the manuscript, the inclusion of the standard k-$\varepsilon$ turbulence model was briefly mentioned in lines 88-90. As it is definitely relevant to stress the importance of turbulence modelling, we will expand more on the inclusion of the turbulence model in a new version of the manuscript. The following new paragraph will replace the sentences about the horizontal and vertical viscosity: "In this RANS model, turbulence is modelled with the standard k-$\epsilon$ model (Launder and Spalding, 1974). The modelled eddy viscosity is added to the molecular viscosity, yielding a non-uniform vertical viscosity $\nu_v$. For the calculations in this article, the horizontal kinematic viscosity $\nu_h$ is set uniform to its molecular value ($\sim 10^{-6} m^2 s^{-1}$). "

*3. The sensitivity of the numerical results on grid should also be discussed. Since the numerical diffusion would contaminate the physics.*

The referee raises an important issue here, although grid sensitivity is usually more an issue in DNS models. A sensitivity analysis is therefore beyond the scope of this paper. We refer to our answer to Referee #3 for a discussion based on some results for grid sensitivity tests. In the paper, we would like to stick to the presentation of the method and show several test cases to verify and validate the model. We recommend a more thorough sensitivity analysis in a future study, as knowledge of the grid sensitivity of the model results is essential for future applications.

In our manuscript, we already focussed on the importance of the model grid selection when discussing the major disadvantage of 3-D models: they are highly computational expensive for the fine meshes required to correctly approach the salt and heat transport. We agree that the modelled physics can be highly influenced by the model grid and that we can better highlight the issue of grid sensitivity in this paper. We therefore decided to add a sentence to the Conclusions that pays attention to the fact that numerical results, and especially those for double-diffusive systems, can be sensitive to

the selection of the model grid. This sentence will be included in a new final paragraph of the Conclusions that sets out the applicability of the model and future recommendations: "Although the model is already able to show expected behaviour in the double-diffusive regime, we recommend a further exploration of its limitations and possibilities. For example, a grid convergence study should indicate whether the selected mesh size yields a convergence of results for all diffusion and advection dominated cases. Further, a comparison with DNS model results would support the validation of the model. In future applications, we stress that this model approach should be employed as a RANS model that simulates thermohaline stratification processes on a larger scale. As such, the model can be favourable in applications that allow an axisymmetric approach."

*4. eq. (12): in 2DV and 3D models, bottom friction is usually accounted for through a bottom roughness. Chezy coefficient is often used in 2DH models. Why do you choose Chezy coefficient instead of bottom roughness? How does this coefficient affect your results?*

We completely agree that the inclusion of a Chézy bottom friction is an unusual approach for a multilayer model. In fact, the presented model code provides the option to calculate with a logarithmic wall approach including the Nikuradse roughness height to determine the bottom friction, which is a far more common practice. The bottom friction is incorporated in the presented cases to slightly impede the high flow velocities that can locally occur, and not to approach a specified level of bottom roughness. Due to familiarity and simplicity, the authors had therefore selected a Chézy coefficient. Instead of what was presented in Eq. 12, the Chézy bottom friction was already scaled to the flow profile in the bottom layer and should have actually been presented as follows:

$$\nu_v \frac{\partial u}{\partial z}\bigg|_{z=-d} = \frac{g}{C^2} \cdot U^2 \cdot \frac{u_{k=1}}{|u_{k=1}|} \tag{1}$$

To assess the effect of the Chézy bottom friction compared to the law of the wall, we repeated Case 3 for both bottom friction boundary conditions (Figure 1). For these calculations, we applied a horizontal mesh size of 5 mm and an inflow velocity of 1 mm s$^{-1}$)[1]. The results show how much the Chézy boundary description affects the flow patterns in the model, especially near the grid centre. The improper flow description near the bottom boundary yields an improper friction of the local friction and in the end yields a far more turbulent flow. Figure 1 shows that the Chézy friction causes a lot more turbulent mixing of heat compared to the logarithmic wall description with a Nikuradse roughness height of 0.1 mm. Also for a roughness height of 10 mm (not shown here), the logarithmic wall law yields a steady growth of the bottom layer without a lot of turbulent mixing.

To conclude, we would also like to stress that the application of the law of the wall is the most common practice for multilayer flow modelling. For this reason, and because of the results that we have shown, we recommend that the users of the model follow this approach instead of using the Chézy bottom friction. In a new upload of our dataset, we will disallow the use of other friction coefficients which are intended for depth-averaged calculations for the axisymmetric case in the model code. In the article, we will not mention the possibility to use a Chézy coefficient anymore, and we will formulate the bottom boundary condition for $u$-momentum for our simulations with the logarithmic wall law. We thank the referee for pointing this out.

*5. theta is used for the tangential direction in section 2.1. However, this becomes alpha in section 2.3. Please make it consistent throughout the paper.*

We thank the referee for making us aware of the inconsistent use of theta. We replaced theta in Section 2.1 by alpha, when introducing the cylindrical coordinates. In the

[Figure]
* * *
[1]It should be mentioned here that the result plotted in Figure 8 of the manuscript was actually calculated for an inflow of $2 \cdot 10^{-4}$ m s$^{-1}$, and not $1 \cdot 10^{-3}$ m s$^{-1}$ as was indicated in Table 1 (we will repeat all simulations with a different bottom friction, so this will be corrected in a new manuscript).

updated version of our article, theta is only employed as an implicitness factor in the theta scheme.

**2  Changes to the manuscript**

Based on the comments of the referee, we will apply several changes to the manuscript:

- We will better define the purpose of the model and the article, by modifying and extending:
  - lines 50-53, better explaining that situations of seepage inflows in shallow waters, causing thermohaline stratification, actually exist;
  - lines 63-64, better explaining why we choose for an axisymmetric approach over an 2-DV approach (to better simulate the volumetric inflow of the central seepage source).

- We made clearer that this RANS model does employ a turbulence model (the standard k-$\epsilon$ model), by modifying and extending lines 87-90.

- We have tested for numerical grid convergence and we will add a sentence to the conclusions to focus on the issue of grid sensitivity of the model.

- The simulations will be performed again and uploaded as a new dataset, where:
  - the simulations will now be done with a logarithmic wall approach where bottom friction is determined by a Nikuradse roughness height instead of a Chézy coefficient;
  - the published code will not allow the option anymore to incorporate Chézy, Manning or any other coefficient that applies for depth-averaged calculations, as soon as the axisymmetric option is selected;
  - the results for Case 3 (double-diffusive convection) will now be presented for an

inflow velocity of 0.001 m s$^{-1}$ (in the previous manuscript, we accidentally added the results for a simulation with an inflow velocity 0.0002 m s$^{-1}$, which was not in accordance with Table 1 and made the results less comparable to Case 4).

- The tangential direction of the axes in cylindrical coordinates is now defined as alpha throughout the article.

**References**

De Louw, P., Vandenbohede, A., Werner, A., and Oude Essink, G.: Natural saltwater upconing by preferential groundwater discharge through boils, J. Hydrol., 490, 74–87, 10.1016/j.jhydrol.2013.03.025, 2013.

Hilgersom, K., Van de Giesen, N., De Louw, P., and Zijlema, M.: Three-dimensional dense distributed temperature sensing for measuring layered thermohaline systems, Water Resour. Res., 52, 6656–6670, 10.1002/2016WR019119, 2016.

Launder, B. and Spalding, D.: The numerical computation of turbulent flows, Comput Methods Appl Mech Eng, 3, 269–289, 10.1016/0045-7825(74)90029-2, 1974.
* * *
[Figure]

$t = 30$ min $\qquad$ $t = 60$ min

$C = 100$ m$^{1/2}$ s$^{-1}$

$n = 0.1$ mm

**Fig. 1.** Comparison between bottom friction boundary conditions described by a Chézy coefficient C and the logarithmic wall with a Nikuradse roughness height n at 30 min. and 60 min. after the start of the run

---

## Author Comment (AC2) · 7 Dec 2016

We would like to thank Referee #2 for giving a thorough review of the manuscript and expressing his concerns. From the complete list of comments, the referee seems to be most concerned about the following issues: the axisymmetric assumption for double-diffusive phenomena, the application of a RANS model instead of a DNS model, and the lack of comparison with laboratory or numerical experiments. Part of these concerns may have been caused by the fact that the referee approaches the modelling of double-diffusive phenomena from a completely different perspective compared to our approach: the referee is clearly an expert in DNS modelling studies to double-diffusive phenomena whereas our RANS approach concerns a larger scale. In other words, we are interested in how double-diffusive systems behave on a larger scale (e.g., 'does a double-diffusive convective system evolve?', and 'how does the location

of a sharp interface evolve over time?'), but we are not interested in fine simulations of the smallest-scale perturbations induced by double-diffusion.

In the following, we address the general comments point by point (Section 1), list our reactions to the detailed comments (Section 2), and provide an overview of the changes made to the manuscript based on the comments (Section 3).

**1 General comments and answers**

*1) What is novel in an axisymmetric model? As pointed out also by Referee #1, the main objectives are not clear. Moreover, practical applications are not discussed (see also next comment).*

This article provides a solution to model density-driven flows in an axisymmetric grid setup. One of the intended purposes of the model is the application in circumstances where double-diffusion potentially occurs. Although an axisymmetric modelling approach is not novel for CFD models (lines 41 to 49 mention examples for a variety of fields of research), it has to our knowledge never been applied in hydraulic free-surface models. The reason why an axisymmetric model can be favourable for our intended application is that we are dealing with a central circular inflow at the bottom boundary, where groundwater of a contrasting salinity and temperature enters the surface water. The axisymmetric modelling approach serves here to closely resemble the volumetric inflow of water and constituents. The aim is to simulate the double-diffusive system that develops, but not the locations of convection cells and salt-fingers, which can never be achieved with the selected model type (as pointed out by the referee in the following comments).

The existence of cases with very local inflows and circular water bodies was already mentioned in the Introduction (lines 50-53). Lines 63-64 mentioned that the developed framework is intended for local saline seepage sources. As I understand from the

comments by Referees 1 and 2, these descriptions are too brief and should provide more information on the intended application. We therefore developed lines 63-64 to a separate paragraph, so that our main objective is clearer: "The development of an axisymmetric variation of SWASH falls in line with our research to localized saline water seepage in Dutch polders. To simulate the effect of a local seepage inflow on the temperature profile of the surface water body, a numerical model is required that accounts for sharp density gradients, a free surface and potential double-diffusive processes. The axisymmetric grid set-up aids in correctly representing the volumetric inflow and modelling the flow processes around the local inflow."

*2) The necessary assumption to formulate the 2-DV model is axial symmetry. However, there is no discussion whether such a symmetry exists in real double-diffusive cases. For instance, the axial symmetry implies that salt fingers are not real "fingers", but "circles" that develop around a central location. Is this reasonable? This is probably the major limitation of this work.*

The referee is obviously right that the "salt-fingers" that develop are not real salt-fingers but circles around the centre of the axisymmetric grid. As pointed out before, our and the referees modelling considerations are completely different. It is not our aim to model exact locations of double-diffusive phenomena, but merely the general behaviour of the system: under the given conditions, will a layered system develop? This is a different question from whether we can approach the exact shape of a salt-finger. In the submitted manuscript, we apparently have not stressed this point enough. We will therefore better stress this at the end of the first paragraph, where the topic is introduced: when we inform the reader about the development of an axisymmetric framework, we will add that this is intended to incorporate the larger-scale effects of double-diffusion. The last sentence of the first paragraph (lines 19-21) will be replaced by: "In this article, we present a framework for a quasi 3-D finite volume approach that allows free-surface flow modelling in an axisymmetric grid. The model framework is intended for a shallow water body where salinity and temperature gradients potentially

induce double-diffusive processes. As such, the model intends to simulate larger-scale features of double-diffusion (i.e., interface locations in a stratified system and heat and salt transport)."

*3) The model is not a DNS model, but a standard RANS model with a k-epsilon model. This means that results are dependent on the parameterization of turbulence, and that the model requires calibration and validation. This is an even more demanding issue in double-diffusive phenomena, which are at the transition between laminar and weakly turbulent flows. I believe that a standard k-epsilon model is not suitable for these conditions, so the whole model formulation is questionable. At least, it cannot be sold as a model that does not require calibration.*

The referee here notes that the model is a RANS model. As stressed before, this type of model is used with a different purpose as compared to the DNS models referred to by the referee. We think that the difference in modelling considerations has lead to different insights by the referee on how the model should be set up.

Further, the referee points out his belief that a standard k-$\varepsilon$ model is not suitable for these conditions. We would like to ask the referee to better explain why he has this belief for these applications. In our eyes, the RANS approach requires a turbulence model to approach the effect of eddies, which due to the mesh size are not directly incorporated in the simulations. This is a major difference from DNS models, which do not require such turbulence models.

Turbulence models do require calibration. One of the reasons why we selected the standard k-$\varepsilon$ model is that it has been applied for decades (Launder and Spalding, 1974) and has become the most popular turbulence model. The model's constants have therefore been confirmed in numerous studies. Like with any model, one should always keep a critical view when applying the k-$\varepsilon$ model, but the historical experience with this model supports its apparent effectiveness.
On the other hand, it is not completely true that DNS models do not require any calibration at all. In the case of DNS models, a certain calibration comes back in the assessments of appropriate mesh sizes and the selection of numerical schemes.

*4) No comparison is provided with laboratory and/or numerical experiments. Only qualitative analogies are discussed, apart from the case of the central inflow (which is likely dominated by advection and not double diffusion). The authors should try to validate their results at least against DNS.*

The referee is right that the quantitative case study with the central inflow is not dominated by double-diffusion, as it concerns a system of unconditionally stable layering. The other case studies consider the double-diffusion dominated systems merely qualitatively, but show that the model functions quite well near critical points where the stability regime changes. Considering the purpose of the model (see answers 1 and 2), these results do support the applicability of the model for its purpose. We agree that a comparison with DNS is recommendable to further test the modelling framework, and we added this as a recommendation to our Conclusions section. For the resubmission of our manuscript, we will add more validation cases that test the following based on flux laws published in Carpenter et al. (2012) and Radko and Smith (2012):

- salt and heat flux across a double-diffusive convection interface;

- interface thickness for salt and heat interfaces in the double-diffusive convection regime;

- salt and heat flux across a salt-fingering interface.

In comment 7, the referee suggests a comparison with the Radko articles about thermohaline staircases. These staircases usually have thicknesses of 20 - 50 m and occur in deeper waters. In contrast to the suggested literature, the intended application of our model typically concerns waters of maximum a few meters deep. Trying to apply the

model to deeper waters would a) surpass the aim of the model, b) would make the free-surface approach irrelevant, and c) would require a too large mesh in the vertical to be conveniently modelled within our model framework (as both a larger depth and a high resolution over this depth are required).

The additional simulations will require more time, and we will therefore ask the editor for an extension of the usual time until resubmission.

*5) One of the major advantages of this formulation is the consideration of the free surface. However, I cannot see where this is a crucial aspect in double-diffusive problems. To my knowledge, these phenomena occur in deep water and are typically not influenced by the dynamics of the free surface, so the authors should explain why this characteristic is important.*

This remark underlines the completely different starting points of the referee and us. The referee is right if the context would be the ocean but we disagree that this is the only context, as our study was initiated based on other cases. As pointed out in the introduction, double-diffusive phenomena also occur on smaller scales like boreholes and solar ponds (lines 36-39). Moreover, Hilgersom et al. (2016) have recently shown that double-diffusive phenomena like salt-fingers can also occur in small drainage canals. In such canals, but for example also solar ponds, the inclusion of a free surface is relevant, and potentially even crucial. We therefore present this method specifically for water bodies at these smaller scales.

*6) The formulation contains some errors (see comments below).*

We thank the referee for his critical view and address the comments in our answers to the detailed comments (Section 2).

*7) The literature review is incomplete and, especially for double diffusion in the diffusive regime, outdated. For instance, no reference is given to recent DNS work, both 2D*

*(e.g., Noguchi & Niino, 2010a,b) and 3D (e.g., Kimura & Smyth, 2007; Carpenter et al., 2012, Sommer et al., 2014). Moreover, papers that analyze the thermohaline staircase (e.g., Radko et al., 2014a,b) could be used to find cases to compare with.*

We thank the referee for these suggestions, which indeed form an important addition to the literature review. In the new version of the paper we will extend the literature review and pay more attention to especially also the DNS work (2-D and 3-D) that has been performed in this field of research. The suggested references will be part of this.

As explained in my answer to the fourth general comment, the papers by Radko will not fulfil to validate the model, as it concerns a problem in a deep water body. The papers by Carpenter et al. (2012) and Sommer et al. (2014) provide better comparable cases.

**2 Answers to detailed comments**

- l. 10: With "expected density and diffusivity driven flow and stratification", we referred to the flow resulting from the sharp density gradients, which result from the double-diffusive processes in these systems. We are changing this part of the sentence to "the expected double-diffusive processes and the resulting density-driven flows".

- l. 57-58: We here refer to the selection of a staggered grid. The staggered grid keeps the conservation of momentum and mass intact, and this conservative property is required for a proper salt and heat transport modelling. This is what we referred to with the sentence "the momentum and mass conservative grid setup allows accurate modelling of transport processes". Because this sentence was not clear, we are changing it to: "the staggered grid allows a momentum and mass conservative solution of the governing equations, which is required for accurate salt and heat transport modelling".

- l. 70: Despite the theoretical requirement to calibrate the turbulence model, the model parameters for the standard k-$\varepsilon$ model model have been found consistent in numerous studies since it was first published by Launder and Spalding (1974) (see also our answer to the third general comment in Section 1). Because we are not aiming to promote this method as a method that does not require calibration (in this sense, it is no different from most other RANS models), we will remove this sentence.

- l. 71: See our answer to the fourth general comment (Section 1).

- l. 88-90: In contrast to DNS simulations, the horizontal scales are generally different from the vertical scales in our applications. For these applications, it is assumed that in the horizontal plane less shear will occur compared to the vertical plane. This causes anisotropy, and explains why we select an advanced turbulence model for the vertical eddy viscosity, compared to a constant horizontal viscosity. Due to the relatively fine mesh, we believe that the horizontal viscosity can be approached by its molecular value. Our simulations, which show examples of the intended model applications, also indicate that vertical mixing is generally larger than horizontal mixing.

- l. 108-109: The referee correctly points out that our sentence improperly suggested that the horizontal diffusivity also includes turbulent diffusion. In fact, the same anisotropy in diffusion was assumed in the transport model as was done in the flow model. Therefore we will change this sentence as follows: "To account for vertical turbulent diffusion, $D_v$ is calculated by adding the molecular diffusivity and turbulent diffusivity: $D = D_{mol} + D_{turb}$."

- Eq. 6: We thank the referee of making us aware of the mistake in this equation. The equation will be reformulated to cylindrical coordinates where Q represents the depth and width integrated velocity, as it was actually employed in our framework: $y \cdot \partial\zeta/\partial t + \partial Q/\partial r = 0$, $Q = UHy$

- l. 117: We disagree with the suggested unimportance of molecular heat and salt diffusion rates, because there are cases where the molecular diffusivities are the main drivers of the salt and heat transport. For example in Carpenter et al. (2012), who are referred to by the referee, it is concluded that the major transport mechanism of salt and heat over an interface in the double-diffusive convective regime is molecular diffusion. This specific case will now be mentioned in the extended literature review and this way supports the importance of variable diffusion rates.

- l. 136: We will reformulate the horizontal mass boundary condition as follows: : $\partial cr/\partial r = 0$

- Eq. 13: $\omega$ is the relative vertical velocity. In the new version of the article, we will define the variable as such and refer to the mathematical definition of $\omega$ in Equation 16 of Zijlema and Stelling (2005).

- l. 164: The anti-creepage terms are used to better approach the horizontal diffusive fluxes over the top and bottom cell-interfaces, which are often not horizontal. The terms are derived from a further expansion of the transport equation on a depth-varying vertical grid. As a further explanation and derivation of the anti-creepage terms can be found elsewhere, we have decided that our current explanation in the article suffices.

- Eq. 20, Eq. 21: The units will be included at the introduction of the variables (line 198 for $\alpha_V$ and $\beta_V$, line 203 for $S$ and $T$).

- l. 214-215: For this, we refer to our answer to the first general comment (Section 1). The introduction will now better state the purpose of our model, which makes the relevance of a central inflow clearer to the reader.

- Fig. 5: The asterisk refers to the shape of the marks of the data points that

are defined relative to depth. We will change this part of the caption to: "The cell depths that are defined relative to the local water depth (as marked by *) are..."

- l. 261-262: The numbers that support this hypothesis were already in the paper: for Case 1, the density ratio $R_\varrho = 2.04$, and for Case 2, $R_\varrho = 1.19$. When $R_\varrho$ approaches 1, a higher mass transport is expected, as we explained in lines 212-214. We are changing this sentence to: "Based on the difference in density ratios, the salt-fingers in Case 2 are expected to transport more salt and heat (Section 2.4)."

- l. 266-269: For this, we refer to our answer to the second general comment (Section 1). As explained in our answer, this is not an issue in our opinion.

- Fig. 7: The interface was not yet plotted in this figure. In a newer version we will plot the interface at the times t=0 and t=5400, as the interface indeed moves upward over time (most notably in Case 2). The depth profiles are averages over the complete horizontal domain, which will be mentioned explicitly in the caption.

- l. 271-272, Fig. 8: Because the bottom layer develops from below, the double-diffusive convective layered structure needs to first build up. However, the layer displayed in Figure 8 already displays double-diffusive convective properties at its interface: the bottom layer only has not extended yet to the outer boundary. It has to be noted that the inflow velocity for this specific case was $2 \cdot 10^{-4}$ m/s, and not $1 \cdot 10^{-3}$ m/s, as was mistakenly written in Table 1 (this will be corrected in the new version of the manuscript). Due to this smaller inflow velocity, it took longer for the bottom layer to develop.

- l. 290-291: The sentence before explains that we experienced difficulties to define our inflow parameters so that the flow will be laminar at once. Therefore, the flow was turbulent at first (because of the very shallow bottom layer), and a laminatisation occurred after the bottom layer had further grown.

- l. 297: The method applied is probably misunderstood. The flow was definitely turbulent during the first 6000 s of the model simulation. Afterwards, a laminarisation occured which is visible from the resulting radial expansion of the salinity and temperature interface. The latter case is what we based the analytical benchmark test on, and the numerical results fairly agree (without the 0.5 % turbulent diffusion). Then we wondered if the slight deviation of the numerical results could be explained by the flow not being completely laminar throughout the domain and at each moment after t = 6000s. In that case, the effect of turbulence would be incorporated in the numerical results. For the benchmark, the effect of turbulence would come back in the diffusivity that is then enhanced by the turbulence. Our calculations for a slightly increased diffusivity better fitted the shapes of the curves for the numerical results in Fig. 8. We therefore concluded that the slight deviation of the numerical results from the benchmark *might* be caused by turbulence.

- l. 299: We will be more precise: "the turbulent diffusion was calculated by dividing an assumed kinematic viscosity $\nu = 10^{-6}$ m2s-1 by the Prandtl-Schmidt number (Equations 8 and 9)." Of course, the assumed kinematic viscosity is not turbulent, but the Prandtl-Schmidt number is also not constant (especially for low turbulence values). We therefore do not claim something like that the flow was turbulent 0.5 % of the time. The calculated 'turbulent diffusivities' only serve as a proxy to study whether turbulence could have been of influence here.

- l. 310: For this, we refer to our answer to the first general comment (Section 1). The introduction will now better state the purpose of our model, which makes the relevance of a central inflow clearer to the reader.

- l. 320-321: We disagree with this: our verification methods show an accordance with double-diffusive theory as it comes down to the expected onset of double-diffusive layering and the occurrence of salt-fingers (which is demonstrated by

calculating the density ratios and Turner angles for the simulated systems). As written in our answer to the fourth general comment (Section 1), we do agree that a quantitative validation was still lacking. In a new version of the article, a quantitative validation approach will be included.

- l. 321:  The referee is right that no double-diffusive processes are validated in the article. Such validation will be added in a future submission. On the other hand, the model is intended for density-gradient systems that can either be subject to stable stratification and double-diffusion. For this reason, a validation for a stable stratification is equally relevant. The fact that the comparison with the benchmark was done with a very specific interface definition (35 % of the step change) was not just cherry picking: it was the interface definition that provided the sharpest image of the interface location, where using other fractions of the step change did not provide a real sharp interface shape (and therefore probably compared less to the analytical benchmark).

**3  Changes to the manuscript**

The following lists the changes that we will make to the manuscript based on the referee's comments:

- We will better define the purpose of the model and the article, by modifying and extending:
  - lines 19-21, better explaining that our model focuses on larger-scale features of thermohaline stratified systems and that we do not simulate exact locations and shapes of the salt-fingers;
  - lines 50-53, better explaining that situations of seepage inflows in shallow waters, causing thermohaline stratification, actually exist;

- lines 63-64, better explaining why we choose for an axisymmetric approach over an 2-DV approach (to better simulate the volumetric inflow of the central seepage source).

- We will provide a better validation based on salt and heat transport across a interface for the salt-fingering and double-diffusive convection regime and the apparent thickness of the interfaces for salt and heat in the double-diffusive convection regime.

- We will extend and update the literature review with the suggested references. Moreover, the literature review will make a clear distinction between DNS models (both 2-D and 3-D) and RANS models.

Further, the changes listed in Section 2 were applied based on the detailed comments.

**References**

Carpenter, J., Sommer, T., and Wüest, A.: Simulations of a double-diffusive interface in the diffusive convection regime, J. Fluid Mech., 711, 411–436, 2012.

Hilgersom, K., Van de Giesen, N., De Louw, P., and Zijlema, M.: Three-dimensional dense distributed temperature sensing for measuring layered thermohaline systems, Water Resour. Res., 52, 6656–6670, doi:10.1002/2016WR019119, 2016.

Launder, B. and Spalding, D.: The numerical computation of turbulent flows, Comput Methods Appl Mech Eng, 3, 269–289, doi:10.1016/0045-7825(74)90029-2, 1974.

Radko, T. and Smith, D. P.: Equilibrium transport in double-diffusive convection, J. Fluid Mech., 692, 5–27, 2012.

Zijlema, M. and Stelling, G. S.: Further experiences with computing non-hydrostatic free-surface flows involving water waves, Int. J. Numer. Meth. Fluids, 48, 169–197, doi:10.1002/fld.821, 2005.

---

## Author Comment (AC3) · 7 Dec 2016

We would like to thank Referee #3 for reading our manuscript and expressing his/her thoughts. We thank the referee for the compliments for the paper structure and the model development. The major concern seems to be that the robustness of the model is insufficiently proven and that a comparison with real world data would support the presented framework. In the following, we answer the general comments point by point (Section 1), list our reactions to the detailed and technical comments (Sections 2 and 3), and provide an overview of the changes made to the manuscript based on the comments (Section 4).

[Figure]

**1 General comments and answers**

*1) It would be good to elaborate on use of SWASH vs a completely new model. Was the primary reason to take advantage of computational infrastructure? It seems like this may have been more work than starting fresh and it would be nice to include further details for this design choice.*

We thank the referee for highlighting this point. In our eyes, the extension of SWASH is advantageous for multiple reasons, which we already described in the article. In our approach, we want to show how a normal 2-DV model can be easily extended to an axisymmetric model by adding few terms. Moreover, SWASH has several features that were required for our intended model application (i.e., a local groundwater inflow into a shallow water body, where the groundwater has a different salinity and temperature). These features were: calculation of the free surface, the non-hydrostatic component, the staggered grid (mass and momentum conservation), and easy extendability of the freely available code.

*2) Please justify this choice of method as opposed to alternative methods, e.g., advanced mesh refinement.*

Our study focuses on the development of a framework for an axisymmetric modelling approach for free-surface models. The methods that the referee refers to are advanced techniques that allow models to quicker find an accurate solution. These are very interesting techniques, but the objective of the current article is to present the derived framework.

*3) Please provide additional discussion of applications and uses for this code.*

Based on comments by Referee #2, we concluded that the intended applicability has not sufficiently been explained. We will therefore extend the paragraphs in the Introduction (lines 50–53 and 63–64) that introduced the applications for which we developed

this modelling framework.

**2 Answers to specific comments**

- l. 146: Based on a comment by Referee #2, we will better formulate this in the Introduction, where we list the advantages of SWASH (lines 56–59). Instead of just mentioning that SWASH has a momentum and mass conservative grid setup, we will now write: "the staggered grid allows a momentum and mass conservative solution of the governing equations, which is required for accurate salt and heat transport modelling".

- l. 150–175: In our opinion, the presentation of the derived numerical framework is one of the major objectives of the article. Therefore, we will keep this part of the manuscript in the main text. Furthermore, it is not uncommon in papers in the field of fluid mechanics to present the numerical framework in the text.

- l. 181: The sentence was not correctly formulated: only the horizontal time integration of the transport equations is explicit. This is not expected to cause problems in the solution of the transport equations given the small time steps employed. The horizontal momentum terms are solved with MacCormack's 2nd order predictor-corrector scheme.

- l. 198: We have increased the white space between the dots and the variables in the equation to increase the readability.

- Fig. 4: Although we do not see directly which lines should get a different width to make the figure clearer, we increased the width of the diffusive flux arrows to distinguish them from the water fluxes at the inflow and outflow. We hope that this will meet the expectation of the reviewer.

  The color maps of all color plots will be changed to viridis.
- Model convergence: We have tested for grid sensitivity in the dynamical case of the double-diffusive convective layer that develops from the central inflow in the streambed (Case 3 in the article). Here, we focus on the sensitivity to the horizontal grid size, since this size is expected to be most influential in the axisymmetric approach. We have performed these tests for inflow velocities of $2 \cdot 10^{-4}$ m s$^{-1}$ and $1 \cdot 10^{-3}$ m s$^{-1}$[1], and for horizontal mesh sizes of 2.5, 5, and 10 mm (the former for the period of 1 hour, the latter two for 2 hours).
  For the inflow of $1 \cdot 10^{-3}$ m s$^{-1}$, Figure 1 presents the results for temperature after 1 hour and 2 hours for the different mesh sizes (the salinity profiles show similar results). In this case where advection dominates, the flow near the seepage inflow, no real differences are seen for the different mesh sizes.
  For the inflow of $2 \cdot 10^{-4}$ m/s, something interesting happens (Figure 2). When the bottom layer is still thin, diffusion dominates this case: the larger diffusion of heat warms the boundary layer of the surface water on top of the inflowing groundwater at a larger rate than that salt is transported upwards. This makes the boundary layer locally unstable and leads to a sudden breaking of the developing bottom layer. This effect is seen for a horizontal mesh size of 2.5 mm, but not for larger mesh sizes, and displays a sensitivity to the grid for cases where the effect of diffusion is dominant for a thin layer near the central inflow. These effects are not seen once the bottom layer has grown further.
  Based on these results, we therefore recommend applying a fine mesh near the central inflow in case the model is applied for very small inflows in combination with the development of a very thin (initial) layer.

- Simplifications: This issue was also raised by Referee #2. The article does not aim to model double-diffusive features in detail (e.g., the shape of salt-fingers),
* * *
[1] It should be mentioned here that the result plotted in Figure 8 of the manuscript was actually calculated for an inflow of $2 \cdot 10^{-4}$ m s$^{-1}$, and not $1 \cdot 10^{-3}$ m s$^{-1}$ as was indicated in Table 1 (based on comments by Referee #1, we will repeat the simulations with a different bottom friction, so this will be corrected in a new manuscript).

but rather their main effect on stratication and salt and heat transport on a larger scale. Whether the model is succesful in resembling these patterns on a larger scale is something that needs to be validated, and we agree that the validation was still lacking in the submitted article. To meet the concerns of the referees about the unclear presentation of our purpose (i.e., the larger scale), we will better stress this in the Introduction.

- Returning to the research question in conclusions: We agree that the Conclusions section does not clearly reflect the ultimate intentions of the article: the modelling of a central seepage inflow at the bottom boundary of a surface water body, where contrast in salinity and temperature can lead to the occurrence of double-diffusive phenomena. In a new submission, we will partly rewrite the last two paragraphs of the Conclusions to make the presented conclusions supportive to the ultimate goal. The paragraphs will be replaced by the following:

"For our purpose of studying shallow water bodies, three aspects were important: 1) the inclusion of a free surface, 2) the efficient solution of a circular seepage inflow, which makes the problem three-dimensional, and 3) a proper simulation of density driven flow and double-diffusivity driven salt and heat transport. The former aspect was already fulfilled by employing the SWASH framework.

The second aspect was solved by assuming axisymmetry for the Navier-Stokes equation in cylindrical coordinates. The derived numerical framework is presented as a Cartesian 2-DV description with few additional terms and width compensation factors. Our implementation of these terms in the non-hydrostatic SWASH model demonstrates the opportunity to easily expand a 2-DV model towards the presented quasi 3-D model.

The third aspect was fulfilled by extending SWASH with a new density and diffusivity module. The case studies demonstrate explainable behaviour for density driven flow and double-diffusivity driven salt and heat transport. The formation of convective layers and salt-fingers are in accordance with the theory of double-
diffusivity. A quantitative validation method was presented to evaluate the model's performance for a cold and saline inflow developing a dense water layer near the bottom. For laminar flow conditions, the numerical model showed a similar radial expansion of the bottom layer as expected from analytical results."

**3  Answers to technical corrections**

- l. 64:  *axi-symmetric* is replaced by *axisymmetric*

- Eq. 6:  Equation 6 is split over two lines

- l. 140:  *the* is added between *in* and *tangential direction*

**4  Changes to the manuscript**

Based on the comments of the referee, we will apply the following changes to the manuscript:

- We will better define the purpose of the model and the article, by modifying and extending:
  - lines 50–53, better explaining that situations of seepage inflows in shallow waters, causing thermohaline stratification, actually exist;
  - lines 63–64, better explaining why we choose for an axisymmetric approach over an 2-DV approach (to better simulate the volumetric inflow of the central seepage source).

Further, the changes listed in Sections 2 and 3 were applied based on the specific and technical comments.

[Figure]

[Figure]

**Fig. 1.** Development of a double-diffusive convection layer for an inflow velocity of 1e-3 m/s.

[Figure]

**Fig. 2.** Development of a double-diffusive convection layer for an inflow velocity of 2e-4 m/s.

---

## Author Response (AR2)

**Response to referee**

27th August 2017

We thank the referee for taking the time to carefully read the manuscript once more. We agree that his/her suggestions can improve and better clarify our manuscript. In the following, we answer the major and minor comments point by point (Sections 1 and 2), and provide an overview of the changes made to the manuscript based on these comments (Section 3).

In general, we think that some of the referee's major concerns are caused by the following difference in perception: 3-D DNS models are the best computational tools to fully represent the phenomena of double-diffusion. We completely agree with the referee on this point. However, we would like to stress the following in our answers: due to the high computational burden to perform large-scale simulations of doube-diffusion, we seek to shorten the calculation times. To this end, we developed a method for specific cases where axisymmetry can be assumed. Additionally, we apply a RANS method: this simplification will lead to inferior results to those that are found with a DNS method, but might be sufficient to represent the double-diffusive transport phenomena of our interest (mainly salt and heat transport accross the interface). Although we cannot be conclusive with respect to the latter point based on our current results, we think that this paper constitutes a good basis for future research.

**1  Major remarks**

*1. The authors claim that they solve the "Navier-Stokes equations" (abstract, line 4). Recalling one of my previous comments, I think that they should use a different terminology, "Reynolds-averaged Navier-Stokes (RANS) equation", also in the abstract and not only in the introduction, in order to avoid misunderstanding with DNS models.*

We will specify the equations we are solving as Reynolds-averaged Navier-Stokes throughout the abstract and manuscript.

*2. Eq. 6 is not correct yet: first of all, $dQ/dt$ should read $dQ/dr$ (the error is a copy&paste from Zijlema and Stelling, 2008)...*

*Second, the equation 6 is not integrated over the depth only, but also over a sector of amplitude y1-y0. I do not understand why this is necessary, and in any case the notation is confusing, the variable y is not defined and there is no*

*reference to fig. 3 where a diagram is shown that is absolutely obscure, if not even wrong (e.g., y should be a tangential coordinate, not orthogonal to r).*

*Why not introducing the equation integrated only over the depth? By the way, the kinematic boundary condition at the free surface has to be specified.*

*In fact, the other equations are not presented as integrated over alpha, and only when discretized the "alpha terms" appear (section 2.3).*

We thank the referee for making us aware of the mistake in Eq. 6 and will change $dQ/dt$ to $dQ/dr$. We agree that the introduction of $y$ in the width integration in Eq. 6 is premature, and we will refrain from using it in the introduction of the governing equations.

As there might be some unclarities about the use of $y$ later on in the manuscript, we would like to add the following. The width integration is introduced to guarantee mass and momentum conservation: the width of the pie slice diverges from the center to the edge. The specific case of the width integrated continuity equation is also used in the discretization procedure of the momentum equations, where the continuity equation multiplied by the velocity is subtracted from the momentum equations to arrive at the discretized momentum equations in their final form. In this discretization procedure, we specify the 'pie slice' that constitutes our mesh as very narrow, allowing us to simplify the integration representation of the width integration over $y$ in Cartesian coordinates ($y = \tan(\alpha)r \approx \alpha r$). The consistent use of $y$ throughout the integration procedure and the fact that $y$ falls out of these equations at a later stage justifies this simplification.

Regarding the comment on the kinematic condition, we refer to our answer on the first minor remark of the referee (Section 2).

*3. The sentence "In non-turbulent thermohaline systems, stability largely depends on density gradients and molecular heat and salt diffusion rates, which in turn are highly dependent on temperature and salinity" (l. 145-146) is not entirely correct. While I completely agree with the first part, the second part is an overstatement. Stability "highly" depends on the difference between heat and salt diffusion rates, but not on their dependence on temperature and salinity. Most of the models do not even consider this dependence and assume constant values for the two (different!) molecular diffusion coefficients.*

The referee mentions that most models do not consider this dependence. We are definitely aware of the fact that other studies usually employ constant values of expansion and diffusion coefficients (for example, thermal expansion values of $2.43\text{-}2.78\cdot10^{-4}$ K$^{-1}$ (Sommer et al., 2013; 2014) or $2\cdot10^{-4}$ K$^{-1}$ (Kunze, 2003)). However, we are not convinced by an argument based on the current convention. In the following, we explain why we are not weakening our statement about the temperature and salinity dependencies of diffusion rates and expansion coefficients.

Although temperature variations in density and diffusion will not lead to an order difference of 1 or more, it is easily demonstrated that they can lead to differences of even more than a factor 2. If we look for example at the thermal expansion coefficient $\alpha_V$, we can show that $\alpha_V$ is non-linearly dependent

[Figure]

Figure 1: Thermal expansion coefficient over salinity and temperature

on temperature and salinity, with values easily varying from 0 to $4 \cdot 10^{-4}$ $K^{-1}$ for the ranges of common temperatures and salinities (Figure 1). This figure is based on Equation 19 in the manuscript, which is based on a regression to the temperature and salinity derivatives of the updated Eckart formula by Wright (1997). The Eckart formula is widely accepted as an accurate density formulation in oceanographic modelling.

With these relationships, we can for example consider the case of double-diffusive convection in a shallow water body with a saline inflowing seepage source (a case that is relevant for the current model set-up). Overnight, the top layer of the fresh water body can easily cool down to a few degrees above the freezing point, whereas the seepage source has a constant temperature of about 11 $^{\circ}$C and a salinity that can reach 5 g $l^{-1}$ (de Louw et al., 2013). Table 1 shows how the density gradients are affected when they are analyzed based on expansion coefficients related either of these temperatures and salinities. Here, values of $R_{\rho}$ are almost a factor 3 higher when using the temperatures and salinities in the lower layer compared to those in the upper layer.

In this light, we would like to point out that density gradients $R_{\rho}^{-1}$ below 1 represent gravitationally unstable systems, that values of $1 < R_{\rho}^{-1} < {\sim}2$ are considered as a regime where turbulence can penetrate the interface and affect the salt and heat transport (e.g., Carpenter et al., 2012), and that values of $R_{\rho}^{-1} > {\sim}2$ generally represent more quiescent layered systems of double-diffusive convection. Being wrong in $R_{\rho}$ by a factor 3 can therefore yield a completely different regime, and largely affect the model results.

Likewise, diffusivities vary with temperature and salinity, although not as much as the thermal expansion coefficient (Figure 2). These graphs are based on Equations 10 and 11 from the manuscript, which are quadratic regression on

Table 1: Thermohaline expansion/contraction coefficients and density gradient ratios

|  | Upper layer | Lower layer | Average |
|---|---|---|---|
| $T$ (ºC) | 4 | 11 | 7.5 |
| $S$ (kg m$^{-3}$) | 0.2 | 5 | 2.6 |
| $\alpha_V$ (K$^{-1}$) | $1.19 \cdot 10^{-4}$ | $0.413 \cdot 10^{-4}$ | $0.812 \cdot 10^{-4}$ |
| $\beta_V$ (kg g$^{-1}$) | $7.71 \cdot 10^{-4}$ | $7.87 \cdot 10^{-4}$ | $7.79 \cdot 10^{-4}$ |
| $R_\rho$ (-) | 0.066 | 0.194 | 0.130 |

[Figure]

Figure 2: Diffusivity of temperature (left) and salinity (right) dependencies on temperature and salinity

data presented in the International Critical Tables of Numerical Data, Physics, Chemistry and Technology (Washburn and West, 1933). For the temperatures and salinities presented in Table 1, the diffusivity of salt would still vary from $0.869 \cdot 10^{-9}$ to $1.04 \cdot 10^{-9}$ m$^2$ s$^{-1}$ in the upper and lower layer. When choosing to use either of these diffusivities, the ~20% difference directly copies to the difference in salt flux over the interface of stratified systems, when molecular diffusion dominates the transport across the interface.

*4. I do not understand why the simulation of cases 1 and 2 (section 3.1) was stopped after 2 hours (l. 377).*

*Similar for cases 3 and 4 (section 3.2): "From the current results, we cannot tell whether the system converges after 2 h to a system where the salt flux exceeds the heat flux" (l. 430-431). The fact that the simulations are short makes the authors hypothesize a subsequent behavior, which is not known. Why not running longer simulations?*

The current paper focuses on the presentation of a novel method. Of course, this should be supported by model validation results that confirm whether the

developed method has the potential to work. Although very interesting, the primary focus is not the study of convergence in long-term simulations, which might have to exceed 14 hours to arrive at well-resolved salinity fields (Carpenter et al., 2012). On the other hand, we agree that it is odd that the simulations have unequal lengths.

For this reason, we have run longer simulations for a 6-hour period. Although these new simulations are still not long enough to arrive at stable results, some clear trends can be seen from the results already. We adapted the Results and discussion session to describe these longer-term trends. Again, we stress that these results cannot lead to firm conclusions about the convergence of the model to a consistent well-resolved salinity and temperature field. However, we regard these tendencies (for example, the decreasing flux ratios in the salt-finger case - Figure 12 in the article) as sufficient to touch upon the validity of this model framework for general double-diffusive mechanisms. Future work should test the model with an improved turbulence modelling approach (see our answer to the next question) to report firmer conclusions on the model convergence in longer-term simulations.

*5. The authors replied in the following way to my previous that the "standard" turbulence model cannot be taken as "perfect" in transitional (laminar/turbulent) double-diffusive systems: "Further, the referee points out his belief that a standard k-epsilon model is not suitable for these conditions. We would like to ask the referee to better explain why he has this belief for these applications" (p. 7 of their replies). However, they seem to agree with my comment in the revised manuscript. For instance, "... might indicate that the standard k-epsilon model does not function for systems with high density gradients" (l. 385-386); "The poor performance of the standard k-epsilon model also appears" (l. 392); "confirming that the standard k-epsilon model suppresses the onset of double-diffusive convection" (l. 403-404); "... caused by a defective turbulence modelling for systems of large density gradients" (l. 498). Frankly speaking, I do not understand such a different opinion in their reply to reviewers and in the manuscript.*

We completely agree with the referee on this point, and we are aware that high density gradients suppress the production and dissipation of turbulence in standard turbulence models. We therefore recommend to extend the turbulence model for future applications (for example towards the 'mixture turbulence model', which is implemented in the modelling package FLUENT). Our new Section 3.6 in the article highlights this need for an extended turbulence model even further.

In this paper, we basically examine whether a relatively simple model formulation already fulfills to come to meaningful results. In the past, some authors have already found reasonable results for stratified systems with the standard k-ε model in its original form (Wyrwa, 2003) or with modified parameters (Venayagamoorthy et al., 2003).

This is the motivation for our current approach in the article: the parameterization of the standard k-ε model has been extensively studied, and the parameters found in these studies were reasonably consistent. For this reason,

we use the same parameters as a best estimate for turbulence formulation until a better formulation for the turbulence model has been implemented.

To return to the start of our answer, we indeed think that a modified turbulence model should be implemented in the future. Our current results seem to confirm this need: we already stress our thoughts on this in the article, as was pointed out by the referee in this question.

*6. ) In line with the comment above about the modelling of turbulence fluxes in systems with high density gradients, it is worth recalling that the actual requirements are quite simple: a model is needed that predict no turbulence in stable regions (hence molecular diffusion) and significantly higher convective fluxes in unstable regions. A few models that exploit this kind of simplified behavior have been proposed and, recently, Toffolon et al. (2015) showed how it is even possible to reproduce – by means of an minimal model of that kind – the formation of double-diffusive staircases. This considerations can be used in the discussion at l. 409-417.*

We will stress the simple requirements for modeling turbulent fluxes in unstable regions across the model space when we suggest extending the turbulence model. The paper by Toffolon et al. (2015) underlines in our eyes how simple this approach can be. In our opinion, the method developed in their article leaves opportunities to develop a numerical approximation of the production and dissipation of turbulence in line with the k-ε model approach.

We thank the referee for this suggestion, and refer to the article by Toffolon et al. (2015) in our discussion on the turbulence model.

*7. When analyzing the case of salt fingers (section 3.2), the authors write: "The numerical results . . . confirm that salt-fingers are formed over the interface (Fig. 11)" (l. 419-420). I do not see what they refer to in figure 11.*

*Are they referring to the vertical stripes? In axisymmetric conditions, these are not fingers, they are circles (as I already noted in my previous review).*

We are referring to the vertical stripes, which in axisymmetric terms would be circles. However, the point that these features would have an unnatural shape in axisymmetric conditions is irrelevant in our eyes: whether fingers, circles or features of any shape in tangential direction, the model tends to deal with these salt-fingering conditions by means of upward and downward moving parcels of water from the interface and, accordingly, result in reasonable measures of salt and heat transport across the interfaces.

*8. As a whole, it seems that the quantitative comparison (sections 3.1 and 3.2) is not so satisfactory.*

We completely agree that the current results are not as good as usually obtained by DNS simulations. The seeming contradiction in our answer might be caused by the different perceptions of the intended model application in this article: we are presenting a modelling approach in which we would like to capture the general behaviour of double-diffusive processes with less computational effort. In fact, this is a first step towards such a modelling approach. The model will be further enhanced and tested in the future. For example, we consider

adding a better approach for turbulence modelling later on, which is expected to improve this RANS modelling approach.

The quantitative comparison is currently not supported by the model results, which represent a time span of the model that is within the spin-up time of the model. The new (longer) simulations for the Cases 1 to 4 are still too short (see our answer to point 4). However, trends in the presented metrics for model performance with respect to double-diffusion show a tendency towards the metrics reported for double-diffusive systems. Considering that the model is likely still spinning up, we consider this tendency sufficient.

*9. I do not see why an axisymmetric 2-D model should be presented as "quasi 3-D".*

This modelling approach has been developed for specific cases that are (almost) symmetrical in tangential direction and where variations in radial direction and over depth are of interest. This way, one can simplify a 3-D system to the solution of a 2-D set of equations, allowing to enhance the speed of calculations drastically. Despite this simplification, the model represents systems that extend in three spatial dimensions (from a classical Cartesian point of view).

However, we agree that the term 'quasi 3-D' is vague and multi-interpretable. For this reason, we have chosen to restrict ourselves to calling the model '2-D axisymmetric' throughout the manuscript. In the introduction, we specify that the model is used for 3-D systems that approach axisymmetry.

**2 Minor remarks**

*- At the free surface (l. 153), a kinematic boundary condition is required as well.*

The following kinematic condition will now be introduced in l. 132 of the text, where we present the equation for the free surface (Eq. 6):

$$w|_{z=\zeta} = \partial\zeta/\partial t + u\partial\zeta/\partial r \qquad (1)$$

*- (l. 419), "Case 4": (Tu = 85.0°) not (Tu = 71.2°).*

We thank the reviewer for making us aware of this error, which we will correct in the manuscript.

**3 Changes to the manuscript**

Based on the comments by the referee, we will apply several changes to the manuscript:

- *Navier-Stokes* has been changed to *Reynolds-averaged Navier-Stokes* on locations in the text where this had not been done sofar (i.e., in the Abstract, Section 2.1, and the Conclusions).

- Equation 6: $\mathrm{d}Q/\mathrm{d}t$ has been changed to $\mathrm{d}Q/\mathrm{d}r$.

- Longer simulations are presented for the Cases 1 to 4, which now have an equal length of 6 hours. The Results and discussion section has been updated to discuss these longer simulations, but this has not lead to changes in the Conclusions section.

- Section 3.6 was added to highlight our discussion of the defects in the turbulence model and to discuss potential future improvements. We suggest that an improvement to the turbulence model could for example be based on the minimal model by Toffolon et al. (2015).

- *Quasi-3D* has been changed to *2-D axisymmetric* throughout the manuscript. The introduction specifies that the model is used for 3-D systems which approach axisymmetry.

- The free surface condition is now explicitly introduced in line 132 of the manuscript.

- In Section 3.2, we corrected the Turner angle of Case 4 (*Tu = 85.0°*).

Attached do this document, we have added a marked-up version of our manuscript in which all changes are highlighted.

[revised manuscript text omitted]

---

## Author Response (AR3)

**Response to referee**

24th October 2017

We would like to thank the referee for taking the time to carefully review the manuscript. His/her suggestions have helped us improving the quality of the article, which in our eyes now clearly states the potential and limitations of our approach. Below, we give a point-by-point response to the remaining issues.

*1. "salt-fingers" in an axisymmetric model (section 3.2). As the authors agree in their reply, their "salt-fingers" would appear as "circles" in 3-D. This should be explicitly stated in the paper because it is not the geometrical structure that a reader would expect when thinking at real salt-fingers.*

The previous version of the manuscript already mentioned that the model is intended "to simulate larger-scale features of double-diffusion" (lines 23-24). We have now added an explicit statement in the Results and discussion section that the salt-fingers would apper as circles in 3-D: "The 3-D representation of these fingers would be circles around the centre of the axisymmetric grid, which is in disagreement with the real-world physics. However, the model does represent the upward and downward moving parcels of water from the interface, and consequently results in reasonable measures of salt and heat transport across the interface." (lines 422-425 in the new version of the manuscript).

*2. I appreciate that the authors run longer simulations (i.e., passing from 4 to 6 hours of simulated time). However, as they state in their reply, the new simulations "are still not long enough to arrive at stable results". I am a bit puzzled, then. The advantage of the proposed method should be "to shorten the calculation times" compared to the "high computational burden to perform large-scale simulations of double-diffusion" that is typical of DNS.*
*Which was the computational time to simulate the 6 hours? If the authors are aware that the new simulations are too short, but they did not run longer ones, I guess that the computational burden is still quite demanding. So my question and request: can the authors report on the computational time of their model? Moreover, can they compare it with that of analogous DNS, and briefly discuss the practical advantages of their approach?*

Our 6-hour simulations took on average 4 days, using four cores of a Intel® Core™ i5-4670 Processor (the simulations with a turbulence model up to half a day faster, those without a turbulence model up to half a day slower). More than the computational burden of one longer term simulation, was the computational burden of doing this effort for the eight simulations in the Cases 1 to 4. We have not run DNS simulations. Therefore, we are sorry to say that we are not able to provide a comparison in terms of simulation time. In order to provide a rough comparison, we have tried to find information on simulation times in the cited DNS studies. Unfortunately, these articles do not provide this information. Moreover, a typical DNS seems to require high-perfomance computing or at least more advanced computing power than the desktop simulations in our study. This seems to be confirmed by the literature. For example, Yoshida and Nagashima (2003) mention that "with the development of high-performance computers, the details of fingers and the resulting turbulent motion can now be resolved better", when they introduce the development of DNS studies. The unequal computer power would complicate a fair comparison of simulation times between the presented modelling framework and a DNS study.

For this reason, we have decided to provide a rough comparison based on typical length and time scales in 3-D DNS studies and the time steps and spatial steps as we have applied them in our approach. Based on the Kolmogorov theory, DNS studies require a spatial step in the order of millimeters and a time step in the order of milliseconds. For example, Carpenter et al. (2012) applied a mesh size in the order of millimeters for their 3-D DNS simulations. Translated to our domain size and and simulation period, this yields a 3-D mesh of $N_x \cdot N_y \cdot N_z \approx 28.7 \cdot 10^6 \cdot 700 \approx 19.8 \cdot 10^9$ cells and $N_t = 21.6 \cdot 10^6$ time steps. This allows us to give a rough estimation of the CPU time of our simulations (about 4 days with $N_r \cdot N_z = 600 \cdot 70 = 42000$ cells and $N_t = 10.8 \cdot 10^6$ time steps) with the time that would be required if the typical DNS simulations were run on the same CPU:

$$T = 4 \cdot \frac{28.7 \cdot 10^6 \cdot 700 \cdot 21.6 \cdot 10^6}{600 \cdot 70 \cdot 10.8 \cdot 10^6} = 3.77 \cdot 10^6$$

This would require 3.77 million days, or over 10,000 years. This calculation confirms that high-performance computing is needed to finish the DNS simulation of such a problem in an acceptable time.

The possibility to run the simulations on a desktop PC is one of the practical advantages of our approach. Of course, one needs to keep in mind here that the model does not resolve all scales of double-diffusion. However, our approach can be advantageous as a pragmatic solution tool for real-world applications where the model domain can be approached by axisymmetry. We have made this explicit in lines 91-93 of the new manuscript: "Rather than a fine-scale model that resolves all the relevant scales for double-diffusive processes, the current approach should be regarded as a pragmatic solution tool that can be run on relatively simple computer infrastructure".

[revised manuscript text omitted]